# Toward Global Convergence of Gradient EM for Over-Parameterized Gaussian Mixture Models

**Weihang Xu**
University of Washington
xuwh@cs.washington.edu

**Maryam Fazel**
University of Washington
mfazel@uw.edu

**Simon S. Du**
University of Washington
ssdu@cs.washington.edu

## Abstract

We study the gradient Expectation-Maximization (EM) algorithm for Gaussian Mixture Models (GMM) in the over-parameterized setting, where a general GMM with $n > 1$ components learns from data that are generated by a single ground truth Gaussian distribution. While results for the special case of 2-Gaussian mixtures are well-known, a general global convergence analysis for arbitrary $n$ remains unresolved and faces several new technical barriers since the convergence becomes sub-linear and non-monotonic. To address these challenges, we construct a novel likelihood-based convergence analysis framework and rigorously prove that gradient EM converges globally with a sublinear rate $O(1/\sqrt{t})$. This is the first global convergence result for Gaussian mixtures with more than 2 components. The sublinear convergence rate is due to the algorithmic nature of learning over-parameterized GMM with gradient EM. We also identify a new emerging technical challenge for learning general over-parameterized GMM: the existence of bad local regions that can trap gradient EM for an exponential number of steps.

## 1 Introduction

Learning Gaussian Mixture Models (GMM) is a fundamental problem in machine learning with broad applications. In this problem, data generated from a mixture of $n \geq 2$ ground truth Gaussians are observed without the label (the index of component Gaussian that data is sampled from), and the goal is to retrieve the maximum likelihood estimation of Gaussian components. The Expectation Maximization (EM) algorithm is arguably the most widely-used algorithm for this problem. Each iteration of the EM algorithm consists of two steps. In the expectation (E) step, it computes the posterior probability of unobserved mixture membership label according to the current parameterized model. In the maximization (M) step, it computes the maximizer of the $Q$ function, which is the likelihood with respect to posterior estimation of the hidden label computed in the E step.

Gradient EM, as a popular variant of EM, is often used in practice when the maximization step of EM is costly or even intractable. It replaces the M step of EM with taking one gradient step on the $Q$ function. Learning Gaussian Mixture Models with EM/gradient EM is an important and widely-studied problem. Starting from the seminal work [Balakrishnan et al., 2014], a flurry of work Daskalakis et al. [2017], Xu et al. [2016], Dwivedi et al. [2018a], Kwon and Caramanis [2020], Dwivedi et al. [2019] have studied the convergence guarantee for EM/gradient EM in various settings. However, these works either only prove local convergence, or consider the special case of 2-Gaussian mixtures. A general global convergence analysis of EM/gradient EM on $n$-Gaussian mixtures still remains unresolved. Jin et al. [2016] is a notable negative result in this regard, where the authors show that on GMM with $n \geq 3$ components, randomly initialized EM will get trapped in a spurious local minimum with high probability.

**Over-parameterized Gaussian Mixture Models.** Motivated by the negative results, a line of work considers the over-parameterized setting where the model uses more Gaussian components than

38th Conference on Neural Information Processing Systems (NeurIPS 2024).

the ground truth GMM, in the hope that it might help the global convergence of EM and bypass the negative result. In such over-parameterized regime, the best that people know so far is from [Dwivedi et al., 2018b]. This work proves global convergence of 2-Gaussian mixtures on one single Gaussian ground truth. The authors also show that EM has a unique sub-linear convergence rate in this over-parameterized setting (compared with the linear convergence rate in the exact-parameterized setting [Balakrishnan et al., 2014]). This motivates the following natural open question:

*Can we prove global convergence of the EM/gradient EM algorithm on general $n$-Gaussian mixtures in the over-parameterized regime?*

In this paper, we take a significant step towards answering this question. Our main contributions can be summarized as follows:

- We prove global convergence of the gradient EM algorithm for learning general $n$-component GMM on one single ground truth Gaussian distribution. This is, to the best of our knowledge, the first global convergence proof for general $n$-component GMM. Our convergence rate is sub-linear, reflecting an inherent nature of over-parameterized GMM (see Remark 3 for details).

- We propose a new analysis framework that utilizes the likelihood function for proving convergence of gradient EM. Our new framework tackles several emerging technical barriers for global analysis of general GMM.

- We also identify a new geometric property of gradient EM for learning general $n$-component GMM: There exists bad initialization regions that traps gradient EM for exponentially long, resulting in an inevitable exponential factor in the convergence rate of gradient EM.

## 1.1 Gaussian Mixture Model (GMM)

We consider the canonical Gaussian Mixture Models with weights $\boldsymbol{\pi} = (\pi_1, \ldots, \pi_n)$ ($\sum_{i=1}^n \pi_i = 1$), means $\boldsymbol{\mu} = (\mu_1^\top, \ldots, \mu_n^\top)^\top$ and unit covariance matrices $I_d$ in $d$-dimensional space. Following a widely-studied setting [Balakrishnan et al., 2014, Yan et al., 2017, Daskalakis et al., 2017], we set the weights $\boldsymbol{\pi}$ and covariances $I_d$ in student GMM as fixed, and the means $\boldsymbol{\mu} = (\mu_1^\top, \ldots, \mu_n^\top)^\top$ as trainable parameters. We use GMM($\boldsymbol{\mu}$) to denote the GMM model parameterized by $\boldsymbol{\mu}$, which can be described with probability density function (PDF) $p_{\boldsymbol{\mu}} : \mathbf{R}^d \to \mathbf{R}_{\geq 0}$ as

$$p_{\boldsymbol{\mu}}(x) = \sum_{i \in [n]} \pi_i \phi(x | \mu_i, I_d) = \sum_{i \in [n]} \pi_i (2\pi)^{-d/2} \exp\left(-\frac{\|x - \mu_i\|^2}{2}\right), \qquad (1)$$

where $\phi(\cdot | \mu, \Sigma)$ is the PDF of $\mathcal{N}(\mu, \Sigma)$, $\pi_1 + \cdots + \pi_n = 1$, $\pi_i > 0, \forall i \in [n]$.

## 1.2 Gradient EM algorithm

The EM algorithm is one of the most popular algorithms for retrieving the maximum likelihood estimator (MLE) on latent variable models. In general, EM and gradient EM address the following problem: given a joint distribution $p_{\boldsymbol{\mu}^*}(x, y)$ of random variables $x, y$ parameterized by $\boldsymbol{\mu}^*$, observing only the distribution of $x$, but not the latent variable $y$, the goal of EM and gradient EM is to retrieve the maximum likelihood estimator

$$\hat{\boldsymbol{\mu}}_{\text{MLE}} \in \arg\max_{\boldsymbol{\mu}} \log p_{\boldsymbol{\mu}}(x).$$

The focus of this paper is the non-convex optimization analysis, so we consider using *population gradient EM* algorithm to learn GMM (1), where the observed variable is $x \in \mathbf{R}^d$ and latent variable is the index of membership Gaussian in GMM. We follow the standard teacher-student setting where a student model GMM($\boldsymbol{\mu}$) with $n \geq 2$ Gaussian components learns from data generated from a ground truth teacher model GMM($\boldsymbol{\mu}^*$). We consider the over-parameterized setting where the ground truth model GMM($\boldsymbol{\mu}^*$) is a single Gaussian distribution $\mathcal{N}(\mu^*, I_d)$, namely $\boldsymbol{\mu}^* = (\mu^{*\top}, \ldots, \mu^{*\top})^\top$. We can then further assume $w.l.o.g.$ that $\mu^* = 0$. Our problem could be seen as a strict generalization of Dwivedi et al. [2018b], where they studied using mixture model of *two Gaussians* with symmetric means (they set constraint $\mu_2 = -\mu_1$) to learn one single Gaussian.

At time step $t = 0, 1, 2, \ldots$, given with parameters $\boldsymbol{\mu}(t) = (\mu_1(t)^\top, \ldots, \mu_n(t)^\top)^\top$, population gradient EM updates $\boldsymbol{\mu}$ via the following two steps

- E step: for each $i \in [n]$, compute the membership weight function $\psi_i : \mathbf{R}^d \to \mathbf{R}$ defined as

$$\psi_i(x|\boldsymbol{\mu}(t)) = \Pr[i|x] = \frac{\pi_i \exp\left(-\frac{\|x - \mu_i(t)\|^2}{2}\right)}{\sum_{k \in [n]} \pi_k \exp\left(-\frac{\|x - \mu_k(t)\|^2}{2}\right)}. \tag{2}$$

- M step: Define $Q(\cdot|, \mu(t))$ as

$$Q(\boldsymbol{\mu}|\boldsymbol{\mu}(t)) = \mathbf{E}_{x \sim \mathcal{N}(0, I_d)}\left[\sum_{i=1}^{n} -\psi_i(x|\boldsymbol{\mu}(t))\frac{\|x - \mu_i\|^2}{2}\right],$$

Gradient EM with step size $\eta > 0$ performs the following update:

$$\mu_i(t+1) = \mu_i(t) - \eta \nabla_{\mu_i} Q(\boldsymbol{\mu}(t)|\boldsymbol{\mu}(t)) = \mu_i(t) - \eta \mathbf{E}_{x \sim \mathcal{N}(0, I_d)}\left[\psi_i(x|\boldsymbol{\mu}(t))(\mu_i(t) - x)\right]. \tag{3}$$

The membership weight function $x \to \psi_i(x|\boldsymbol{\mu})$ represents the posterior probability of data point $x$ being sampled from the $i^{\text{th}}$ Gaussian of GMM$(\boldsymbol{\mu})$. For ease of notation, we sometimes simply write $\psi_i(x|\boldsymbol{\mu})$ as $\psi_i(x)$ when the choice of $\boldsymbol{\mu}$ is obvious.

## 1.3 Loss function of gradient EM

Since the task of gradient EM is to find the MLE over ground truth distribution $p_{\boldsymbol{\mu}^*}$, we can define the MLE loss function for gradient EM as

$$\mathcal{L}(\boldsymbol{\mu}) = D_{\text{KL}}(p_{\boldsymbol{\mu}^*}||p_{\boldsymbol{\mu}}) = -\mathbf{E}_{x \sim p_{\boldsymbol{\mu}^*}}\left[\log\left(\frac{p_{\boldsymbol{\mu}}(x)}{p_{\boldsymbol{\mu}^*}(x)}\right)\right]. \tag{4}$$

The loss $\mathcal{L}$ is the Kullback–Leibler (KL) divergence between the ground truth GMM and the student model GMM. Since finding MLE is equivalent to minimizing the KL divergence between model and the ground truth, the goal of gradient EM is equivalent to finding the global minimum of loss $\mathcal{L}$. In other words, proving that gradient EM finds the MLE is equivalent with proving the convergence of $\mathcal{L}$ to 0. However, we are going to present another reason why loss function $\mathcal{L}$ is important, for it is also closely related to the dynamics of gradient EM.

**Gradient EM is gradient descent on $\mathcal{L}$.** We present the following important observation. The proof is deferred to appendix.

**Fact 1.** *For any $\boldsymbol{\mu}$, $\nabla Q(\boldsymbol{\mu}|\boldsymbol{\mu}) = \nabla \mathcal{L}(\boldsymbol{\mu})$.*

Fact 1 states that the gradient of $Q$ function that gradient EM optimizes in each iteration is identical to the gradient of loss function $\mathcal{L}$. This observation is very useful since it implies that gradient EM is equivalent to gradient descent (GD) algorithm on $\mathcal{L}$. This observation is not a new discovery of ours but actually a wide-spread folklore (see [Jin et al., 2016]). However, our new contribution is to observe Fact 1 is very helpful for analyzing gradient EM, and to construct a new convergence analysis framework for gradient EM based on it.

## 1.4 Notation

In this paper, we adopt the following notational conventions. We denote $\{1, 2, \ldots, n\}$ with $[n]$. $\boldsymbol{\mu} = (\mu_1^\top, \ldots, \mu_n^\top)^\top \in \mathbf{R}^{nd}$ denotes the parameter vector of GMM obtained by concatenating Gaussian mean vectors $\mu_1, \ldots, \mu_n$ together. For any vector $\mu$, $\mu(t)$ denotes its value at time step $t$, sometimes we omit this iteration number $t$ when its choice is clear and simply abbreviate $\mu(t)$ as $\mu$. We define a shorthand of expectation taken over the ground truth GMM $\mathbf{E}_{x \sim \mathcal{N}(0, I_d)}[\cdot]$ as $\mathbf{E}_x[\cdot]$. For any vector $v \neq 0$, we use $\overline{v} := v/\|v\|$ to denote the normalization of $v$. We define (with a slight abuse of notation) $i_{\max} := \arg\max_{i \in [n]}\{\|\mu_i\|\}$ as the index of $\mu_i$ with the maximum norm, and $\mu_{\max} := \|\mu_{i_{\max}}\| = \max_{i \in [n]}\{\|\mu_i\|\}$ as the maximum norm of $\mu_i$. In particular, $\mu_{\max}(t) = \max\{\|\mu_1(t)\|, \ldots, \|\mu_n(t)\|\}$. Similarly, $\pi_{\min} := \min_{i \in [n]} \pi_i$ and $\pi_{\max} := \max_{i \in [n]} \pi_i$ denotes the minimal and maximal $\pi_i$, respectively. We use $\nabla_{\mu_i}\mathcal{L}$ to denote the gradient of $\mu_i$ on $\mathcal{L}$, and $\nabla\mathcal{L} = (\nabla_{\mu_1}\mathcal{L}^\top, \ldots, \nabla_{\mu_n}\mathcal{L}^\top)^\top$ denotes the collection of all gradients. Finally we define a potential function $U : \mathbf{R}^{nd} \to \mathbf{R}$ for GMM$(\boldsymbol{\mu})$ as

$$U(\boldsymbol{\mu}) = \sum_{i \in [n]} \|\mu_i\|^2.$$

## 1.5 Technical overview

Here we provide a brief summary of the major technical barriers for our global convergence analysis and our techniques for overcoming them.

**New likelihood-based analysis framework.** The traditional convergence analysis for EM/gradient EM in previous works Balakrishnan et al. [2014], Yan et al. [2017], Kwon and Caramanis [2020] proceeds by showing the distance between the model and the ground truth GMM in the *parameter space* contracts linearly in every iteration. This type of approach meets new challenges in the over-parameterized $n$-Gaussian mixture setting since the convergence is both sub-linear and non-monotonic. To address these problems, we propose a new likelihood-based convergence analysis framework: instead of proving the convergence of parameters, our analysis proceeds by showing the likelihood loss function $\mathcal{L}$ converges to $0$. The new analysis framework is more flexible and allows us to overcome the aforementioned technical barriers.

**Gradient lower bound.** The first step of our global convergence analysis constructs a gradient lower bound. Using some algebraic transformation techniques, we convert the gradient projection $\langle \mathcal{L}(\boldsymbol{\mu}), \boldsymbol{\mu} \rangle$ into the expected norm square of a random vector $\tilde{\psi}(x)$. (See Section (4) for the full definition). Although lower bounding the expectation of $\tilde{\psi}$ is very challenging, our key idea is that the gradient of $\tilde{\psi}$ has very nice properties and can be easily lower bounded, allowing us to establish the gradient lower bound.

**Local smoothness and regularity condition.** After obtaining the gradient lower bound, the missing component of the proof is a smoothness condition of the loss function $\mathcal{L}$. Since proving the smoothness of $\mathcal{L}$ is hard in general, we define and prove a weaker notion of local smoothness, which suffices to prove our result. In addition, we design and use an auxiliary function $U$ to show that gradient EM trajectory satisfies the locality required by our smoothness lemma.

## 2 Related work

### 2.1 2-Gaussian mixtures

There is a vast literature studying the convergence of EM/gradient EM on 2-component GMM. The initial batch of results proves convergence within a infinitesimally small local region [Xu and Jordan, 1996, Ma et al., 2000]. Balakrishnan et al. [2014] proves for the first time convergence of EM and gradient EM within a non-infinitesimal local region. Among the later works on the same problem, Klusowski and Brinda [2016] improves the basin of convergence guarantee, Daskalakis et al. [2017], Xu et al. [2016] proves the global convergence for 2-Gaussian mixtures. These works focused on the exact-parameterization scenario where the number of student mixtures is the same as that of the ground truth. More recently, Wu and Zhou [2019] proves global convergence of 2-component GMM without any separation condition. Their result can be viewed as a convergence result in the over-parameterized setting where the student model has two Gaussians and the ground truth is a single Gaussian. On the other hand, their setting is more restricted than ours because they require the means of two Gaussians in the student model to be symmetric around the ground truth mean. Weinberger and Bresler [2021] extends the convergence guarantee to the case of unbalanced weights. Another line of work Dwivedi et al. [2018b, 2019, 2018a] studies the over-parameterized setting of using 2-Gaussian mixture to learn a single Gaussian and proves global convergence of EM. Our result extends this type of analysis to the general case of $n$-Gaussian mixtures, which requires significantly different techniques. We note that going beyond Gaussian mixture models, there are also works studying EM algorithms for other mixture models such as a mixture of linear regression Kwon et al. [2019].

### 2.2 N-Gaussian mixtures

Another line of results focuses on the general case of $n$ Gaussian mixtures. Jin et al. [2016] provides a counter-example showing that EM does not converge globally for $n > 2$ (in the exact-parameterized case). Dasgupta and Schulman [2000] prove that a variant of EM converges to MLE in two rounds for $n$-GMM. Their result relies on a modification of the EM algorithm and is not comparable with ours. [Chen et al., 2023] analyzes the structure of local minima in the likelihood function of GMM. However, their result is purely geometric and does not provide any convergence guarantee.

A series of paper Yan et al. [2017], Zhao et al. [2018], Kwon and Caramanis [2020], Segol and Nadler follow the framework proposed by Balakrishnan et al. [2014] to prove the *local* convergence of EM for $n$-GMM. While their result applies to the more general $n$-Gaussian mixture ground truth setting, their framework only provides local convergence guarantee and cannot be directly applied to our setting.

## 2.3 Slowdown due to over-parameterization

This paper gives an $O\left(1/\sqrt{t}\right)$ bound for fitting over-parameterized Gaussian mixture models to a single Gaussian. Recall that to learn a single Gaussian, if one's student model is also a single Gaussian, then one can obtain an $\exp(-\Omega(t))$ rate because the loss is strongly convex. This slowdown effect due to over-parameterization has been observed for Gaussian mixtures in Dwivedi et al. [2018a], Wu and Zhou [2019], but has also been observed in other learning problems, such as learning a two-layer neural network Xu and Du [2023], Richert et al. [2022] and matrix sensing problems [Xiong et al., 2023, Zhang et al., 2021, Zhuo et al., 2021].

## 3 Main results

In this section, we present our main theoretical result, which consists of two parts: In Section 3.1 we present our global convergence analysis of gradient EM, in Section 3.2 we prove that an exponentially small factor in our convergence bound is inevitable and cannot be removed. All omitted proofs are deferred to the appendix.

## 3.1 Global convergence of gradient EM

We first present our main result, which states that gradient EM converges to MLE globally.

**Theorem 2** (Main result)**.** *Consider training a student $n$-component GMM initialized from $\boldsymbol{\mu}(0) = (\mu_1(0)^\top, \ldots, \mu_n(0)^\top)^\top$ to learn a single-component ground truth GMM $\mathcal{N}(0, I_d)$ with population gradient EM algorithm. If the step size satisfies $\eta \leq O\left(\frac{\exp(-8U(0))\pi_{\min}^2}{n^2 d^2 (\frac{1}{\mu_{\max}(0)} + \mu_{\max}(0)^2)}\right)$, then gradient EM converges globally with rate*

$$\mathcal{L}(\boldsymbol{\mu}(t)) \leq \frac{1}{\sqrt{\gamma t}},$$

*where $\gamma = \Omega\left(\frac{\eta \exp(-16U(0))\pi_{\min}^4}{n^2 d^2 (1 + \mu_{\max}(0)\sqrt{dn})^4}\right) \in \mathbf{R}^+$. Recall that $\mu_{\max}(0) = \max\{\|\mu_1(0)\|, \ldots, \|\mu_n(0)\|\}$ and $U(0) = \sum_{i \in [n]} \|\mu_i(0)\|^2$ are two initialization constants.*

**Remark 3.** *Without over-parameterization, for learning a single Gaussian, one can obtain a linear convergence $\exp(-\Omega(t))$. We would like to note that the sub-linear convergence rate guarantee of gradient EM stated in Theorem 2 ($\mathcal{L}(\boldsymbol{\mu}(t)) \leq O(1/\sqrt{t})$) is due to the inherent nature of the algorithm. Dwivedi et al. [2018b] studied the special case of using 2 Gaussian mixtures with symmetric means to learn a single Gaussian and proved that EM has sublinear convergence rate when the weights $\pi_i$ are equal. Since Theorem 2 studies the more general case of $n$ Gaussian mixtures, this type of subexponential convergence rate is the best than we can hope for.*

**Remark 4.** *The convergence rate in Theorem 2 has a factor exponentially small in the initialization scale ($\gamma \propto \exp(-16U(0))$). We would like to stress that this is again due to algorithmic nature of the problem rather than the limitation of analysis. In Section 3.2, we prove that there exists bad regions with exponentially small gradients so that when initialized from such region, gradient EM gets trapped locally for $\exp(\Omega(U(0)))$ number of steps. Therefore, a convergence speed guarantee exponentially small in $U(0)$ is inevitable and cannot be improved.*

**Remark 5.** *Theorem 2 is fundamentally different from convergence analysis for EM/gradient EM in previous works Yan et al. [2017], Dwivedi et al. [2019], Balakrishnan et al. [2014] which proved monotonic linear contraction of parameter distance $\|\boldsymbol{\mu}(t) - \boldsymbol{\mu}^*\|$. But our result also implies global convergence since loss function $\mathcal{L}$ converging to 0 is equivalent to convergence of gradient EM to MLE.*

**Remark 6.** *The convergence result in Theorem 2 is for population gradient EM, but it also implies global convergence for sample-based gradient EM as the sample size tends to infinity. For a similar reduction from population EM to sample EM, see Section 2.2 of [Xu et al., 2016].*

## 3.2 Necessity of exponentially small factor in convergence rate

In this section we prove that a factor exponentially small in initialization scale $(\exp(-\Theta(U(0))))$ is inevitable in the global convergence rate guarantee of gradient EM. Particularly, we show the existence of bad regions such that initialization from this region traps gradient EM for exponentially long time before final convergence. Our result is the following theorem.

**Theorem 7** (Existence of bad initialization region). *For any* $n \geq 3$, *define* $\tilde{\boldsymbol{\mu}}(0) = (\mu_1^\top(0), \ldots, \mu_n^\top(0))$ *as follows:* $\mu_1(0) = 12\sqrt{d}e_1, \mu_2(0) = -12\sqrt{d}e_1, \mu_3(0) = \cdots = \mu_n(0) = 0$, *where* $e_1$ *is a standard unit vector. Then population gradient EM initialized with means* $\tilde{\boldsymbol{\mu}}(0)$ *and equal weights* $\pi_1 = \ldots = \pi_n = 1/n$ *will be trapped in a bad local region around* $\tilde{\boldsymbol{\mu}}(0)$ *for exponentially long time*

$$T := \frac{1}{30\eta}e^d = \frac{1}{30\eta}\exp(\Theta(U(0))).$$

*More rigorously, for any* $0 \leq t \leq T, \exists i \in [n]$ *such that*

$$\|\mu_i(t)\| \geq 10\sqrt{d}.$$

Theorem 7 states that, when initialized from some bad points $\boldsymbol{\mu}(0)$, after $\exp(\Theta(U(0)))$ number of time steps, gradient EM will still stay in this local region and remain $10\sqrt{d}$ distance away from the global minimum $\boldsymbol{\mu} = 0$. Therefore an exponentially small factor in convergence rate is inevitable.

**Remark 8.** *Theorem 7 eliminates the possibility of proving any polynomial convergence rate of gradient EM from arbitrary initialization. However, it is still possible to prove that, with some specific smart initialization schemes, gradient EM avoids the bad regions stated in Theorem 7 and enjoys a polynomial convergence rate. We leave this as an interesting open question for future analysis.*

## 4 Proof overview

In this section, we provide a technical overview of the proof in our main result (Theorem 2 and Theorem 7).

### 4.1 Difficulties of a global convergence proof and our new analysis framework

Proving the global convergence of gradient EM for general $n$-Gaussian mixture is highly nontrivial. While there have been many previous works [Balakrishnan et al., 2014, Yan et al., 2017, Dwivedi et al., 2018b] studying either local convergence or the special case of 2-Gaussian mixtures, they all focus on showing the contraction of parametric error. Namely, their proof proceeds by showing the distance between the model parameter and the ground truth contracts, usually by a fixed linear ratio, in each iteration of the algorithm. However, this kind of approach faces various challenges for our general problem where the convergence is both *sublinear* and *non-monotonic*. Since the convergence rate is sublinear (see Remark 3), showing a linear contraction per iteration is no longer possible. Since the convergence is non-monotonic[1], we also cannot show a strictly decreasing parametric distance.

To address these challenges, we propose a new convergence analysis framework for gradient EM by proving the convergence of *likelihood* $\mathcal{L}$ instead of the convergence of parameters $\boldsymbol{\mu}$. There are several benefits for considering the convergence from the perspective of MLE loss $\mathcal{L}$. Firstly, it naturally addresses the problem of non-monotonic and sub-linear convergence since we only need to show $\mathcal{L}$ decreases as the algorithm updates. Also, since gradient EM is equivalent with running gradient descent on loss function $\mathcal{L}$ (see Section 1.3), we can apply techniques from the optimization theory of gradient descent to facilitate our analysis.

### 4.2 Proof ideas for Theorem 2

We first briefly outline our proof of Theorem 2.

**Proof roadmap.** Our proof of Theorem 2 consists of three steps. Firstly, we prove a gradient lower bound for $\mathcal{L}$ (Theorem 12). Then we prove that the MLE $\mathcal{L}$ is *locally smooth* (Theorem 13). Finally,

---

[1]To see this, consider $n = 2, \mu_1 = 0, \mu_2 = (1, 0, \ldots, 0)^\top$, then the norm of $\mu_1$ strictly increases after one iteration.

we combine the gradient lower bound and the smoothness condition to prove the global convergence of $\mathcal{L}$ with mathematical induction.

**Step 1: Gradient lower bound.**

Our first step aims to show that the gradient norm of $\mathcal{L}(\boldsymbol{\mu})$ is lower bounded by the distance of $\boldsymbol{\mu}$ to the ground truth. To do this, we need a few preliminary results. Inspired by Chen et al. [2023], we use Stein's identity [Stein, 1981] to perform an algebraic transformation of the gradient. Recalling the definition of $\psi_i$ in (2), we have the following lemma.

**Lemma 9.** *For any GMM($\boldsymbol{\mu}$), $i \in [n]$, the gradient of $Q$ satisfies*

$$\nabla_{\mu_i} \mathcal{L}(\boldsymbol{\mu}) = \nabla_{\mu_i} Q(\boldsymbol{\mu}|\boldsymbol{\mu}) = \mathbf{E}_x \left[ \psi_i(x) \sum_{k \in [n]} \psi_k(x) \mu_k \right].$$

The gradient expression above is equivalent with the form in (3), but is easier to manipulate. Using the transformed gradient in Lemma 9, we have the following corollary.

**Corollary 10.** *Define vector $\tilde{\psi}_{\boldsymbol{\mu}}(x) := \sum_{i \in [n]} \psi_i(x) \mu_i$. For any GMM($\boldsymbol{\mu}$), the projection of the gradient of $\nabla \mathcal{L}(\boldsymbol{\mu})$ onto $\boldsymbol{\mu}$ satisfies*

$$\langle \nabla \mathcal{L}(\boldsymbol{\mu}), \boldsymbol{\mu} \rangle = \langle \nabla_{\boldsymbol{\mu}} Q(\boldsymbol{\mu}|\boldsymbol{\mu}), \boldsymbol{\mu} \rangle = \sum_{i \in [n]} \langle \nabla_{\mu_i} Q(\boldsymbol{\mu}|\boldsymbol{\mu}), \mu_i \rangle = \mathbf{E}_x \left[ \left\| \tilde{\psi}_{\boldsymbol{\mu}}(x) \right\|^2 \right].$$

Corollary 9 is important since it converts the projection of gradient $\nabla \mathcal{L}(\boldsymbol{\mu})$ onto $\boldsymbol{\mu}$ to the expected norm square of a vector $\tilde{\psi}_{\boldsymbol{\mu}}$. Since a lower bound of the gradient projection implies a lower bound of the gradient, we only need to construct a lower bound for $\langle \nabla \mathcal{L}(\boldsymbol{\mu}), \boldsymbol{\mu} \rangle = \mathbf{E}_x \left[ \left\| \tilde{\psi}_{\boldsymbol{\mu}}(x) \right\|^2 \right]$. Since $\left\| \tilde{\psi}_{\boldsymbol{\mu}}(x) \right\|^2$ is always non-negative, we already know that the gradient projection is non-negative. But lower bounding $\mathbf{E}_x \left[ \left\| \tilde{\psi}_{\boldsymbol{\mu}}(x) \right\|^2 \right]$ is still highly nontrivial since the expression of $\tilde{\psi}$ is complicated and hard to handle. However, our key observation is that, *although $\tilde{\psi}$ itself is hard to bound, its gradient has nice properties and can be handled gracefully*:

$$\nabla_x \tilde{\psi}_{\boldsymbol{\mu}}(x) = \frac{1}{2} \sum_{i,j \in [n]} \psi_i(x) \psi_j(x) (\mu_i - \mu_j)(\mu_i - \mu_j)^\top. \tag{5}$$

The gradient (5) is nicely-behaved. One can see immediately from (5) that the matrix $\nabla_x \tilde{\psi}_{\boldsymbol{\mu}}(x)$ is positive-semi-definite, and its eigenvalues can be directly bounded. To utilize these properties, we use the following algebraic trick to convert the task of lower bounding $\tilde{\psi}$ itself into the task of lower bounding its gradient.

$$\mathbf{E}_x \left[ \| \tilde{\psi}_{\boldsymbol{\mu}}(x) \|^2 \right] = \frac{1}{4} \mathbf{E}_x \left[ \left( \int_{t=-1}^{1} \|x\| \cdot \overline{x}^\top \nabla \tilde{\psi}_{\boldsymbol{\mu}}(tx) \overline{x} \mathrm{d}t \right)^2 \right]. \tag{6}$$

Recall that $\overline{x} = \frac{x}{\|x\|}$. See detailed derivation in (23). Using (5), combined with the properties of $\nabla_x \tilde{\psi}_{\boldsymbol{\mu}}(x)$, we can obtain the following lemma (Recall that $U = \sum_{i \in [n]} \|\mu_i\|^2$.):

**Lemma 11.** *For any GMM($\boldsymbol{\mu}$) we have*

$$\mathbf{E}_x \left[ \| \tilde{\psi}_{\boldsymbol{\mu}}(x) \|^2 \right] \geq \frac{\exp(-8U)}{40000 d (1 + 2\mu_{\max} \sqrt{d})^2} \left( \sum_{i,j \in [n]} \pi_i \pi_j \| \mu_i - \mu_j \|^2 \right)^2.$$

On top of Lemma 11, we can easily lower bound the gradient projection in the following lemma, finishing the first step of our proof.

**Lemma 12** (Gradient projection lower bound). *For any GMM($\mu$) we have*

$$\langle \nabla_{\mu} Q(\mu|\mu), \mu \rangle = \mathbf{E}_x[\|\tilde{\psi}_{\mu}(x)\|^2] = \Omega \left( \frac{\exp\left(-8U\right) \pi_{\min}^2}{d(1 + \mu_{\max}\sqrt{d})^2} \mu_{\max}^4 \right).$$

**Step 2: Local smoothness.**

To construct a global convergence analysis for gradient-based methods, after obtaining a gradient lower bound, we still need to prove the smoothness of loss $\mathcal{L}$. (Recall that global smoothness of function $f$ means that there exists constant $C$ such that $\|\nabla f(x_1) - \nabla f(x_2)\| \leq C\|x_1 - x_2\|, \forall x_1, x_2$.) However, proving the smoothness for $\mathcal{L}$ in general is very challenging since the membership function $\psi_i$ cannot be bounded when $\mu$ is unbounded. To address this issue, we prove that $\mathcal{L}$ is locally smooth, *i.e.*, the smoothness between two points $\mu$ and $\mu'$ is satisfied if both $\|\mu\|$ and $\|\mu - \mu'\|$ are upper bounded. Our result is the following theorem.

**Theorem 13** (Local smoothness of loss function). *At any two points $\mu = (\mu_1^\top, \ldots, \mu_n^\top)^\top$ and $\mu + \delta = ((\mu_1 + \delta_1)^\top, \ldots, (\mu_n + \delta_n)^\top)^\top$, if*

$$\|\delta_i\| \leq \frac{1}{\max\{6d, 2\|\mu_i\|\}}, \forall i \in [n],$$

*then the loss function $\mathcal{L}$ satisfies the following smoothness property: for any $i \in [n]$ we have*

$$\|\nabla_{\mu_i + \delta_i} \mathcal{L}(\mu + \delta) - \nabla_{\mu_i} \mathcal{L}(\mu)\| \leq n\mu_{\max}(30\sqrt{d} + 4\mu_{\max})\|\delta_i\| + \sum_{k \in [n]} \|\delta_k\|. \tag{7}$$

**Step 3: putting everything together.**

Given the gradient lower bound and the smoothness condition, we still need to resolve two remaining problems. The first one is that the gradient lower bound in Lemma 12 is given in terms of $\mu$, which we need to convert to a lower bound in terms of $\mathcal{L}(\mu)$. For this we need the following upper bound of $\mathcal{L}$.

**Theorem 14** (Loss function upper bound). *The loss function can be upper bounded as*

$$\mathcal{L}(\mu) \leq \sum_{i \in [n]} \frac{\pi_i}{2}\|\mu_i\|^2 \leq \frac{\mu_{\max}^2}{2}.$$

The second problem is that our local smoothness theorem requires $\mu$ to be bounded, therefore we need to show a regularity condition that for each $i$, $\mu_i(t)$ stays in a bounded region during gradient EM updates. This is not easy to prove for each individual $\mu_i$ due to the same non-monotonic issue mentioned in Section 4.1. To establish such a regularity condition, we use the potential function.$U$ to solve this problem. We prove that $U$ remains bounded along the gradient EM trajectory, implying each $\mu_i$ remains well-behaved. With this regularity condition, combined with the previous two steps, we finish the proof of Theorem 2 via mathematical induction.

### 4.3  Proof ideas for Theorem 7

Proving Theorem 7 is much simpler. The idea is natural: we found that there exists some bad regions where the gradient of $\mathcal{L}$ is exponentially small, characterized by the following lemma.

**Lemma 15** (Gradient norm upper bound). *For any $\mu$ satisfying $\|\mu_1\|, \|\mu_2\| \geq 10\sqrt{d}, \|\mu_3\|, \ldots, \|\mu_n\| \leq \sqrt{d}$, the gradient of $\mathcal{L}$ at $\mu$ can be upper bounded as*

$$\|\nabla_{\mu_i} \mathcal{L}(\mu)\| \leq 2(\|\mu_3\| + \cdots + \|\mu_n\|) + 2\exp(-d)(\|\mu_1\| + \|\mu_2\|), \forall i \in [n].$$

Utilizing Lemma 15, we can prove Theorem 7 by showing that initialization from these bad regions will get trapped in it for exponentially long, since the gradient norm is exponentially small. The full proof can be found in Appendix B.2.

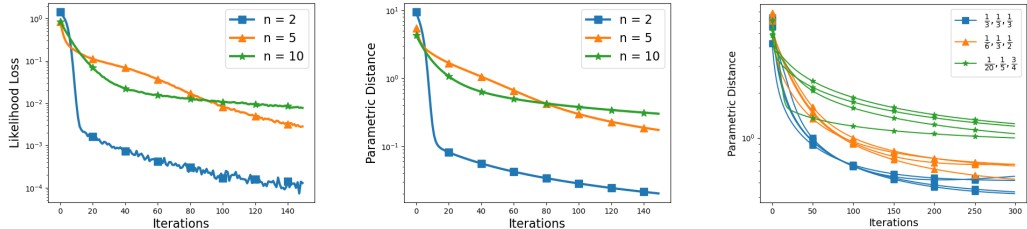

**Figure 1:** Left: Sublinear convergence of the likelihood loss $\mathcal{L}$. Middle: Sublinear convergence of the parametric distance $\sum_{i \in [n]} \pi_i \|\mu_i - \mu^*\|^2$ between student GMM and the ground truth. Right: Impact of different mixing weights on the convergence speed.

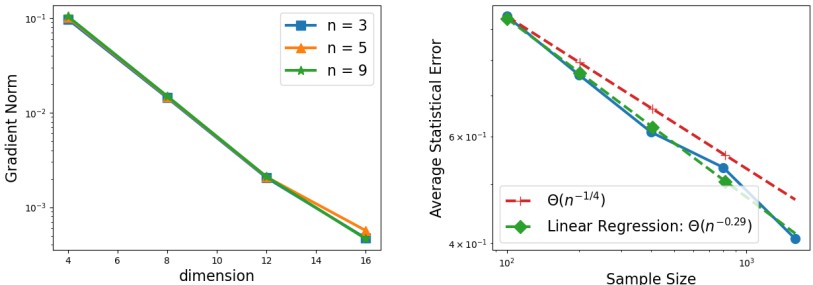

**Figure 2:** Left: Gradient norm $\|\nabla \mathcal{L}(\boldsymbol{\mu}(0))\|$ in the counter-example in Theorem 7 decreases exponentially fast w.r.t. dimension $d$. Right: The statistical error (blue line) approximately scales as $\sim n^{-1/4}$ with sample size $n$.

## 5 Experiments

In this section we experimentally explore the behavior of gradient EM on GMMs.

**Convergence rates.** We choose the experimental setting of $d = 5, \eta = 0.7$. We use $n = 2, 5, 10$ Gaussian mixtures to learn data generated from one single ground truth Gaussian distribution $\mathcal{N}(\mu^*, I_d)$, respectively. Since a closed form expression of the population gradient is intractable, we approximate the gradient step via Monte Carlo method, with sample size $3.5 \times 10^5$. The mixing weights of student GMM are randomly sampled from a standard Dirichlet distribution and set as fixed during gradient EM update. The covariances of all component Gaussians are set as the identity matrix. We recorded the convergence of likelihood function $\mathcal{L}$ (estimated also by Monte Carlo method on fresh samples each iteration) and parametric distance $\sum_{i \in [n]} \pi_i \|\mu_i - \mu^*\|^2$ along gradient EM trajectory. The results are reported in Figure 1 (left and middle panel). Both the likelihood $\mathcal{L}$ and the parametric distance converges sub-linearly.

**Weight configurations.** We train 3-component GMM with 3-different weight configurations and report 4 runs each configuration in Figure 1 (right). Blue: $(\frac{1}{3}, \frac{1}{3}, \frac{1}{3})$. Orange: $(\frac{1}{6}, \frac{1}{3}, \frac{1}{2})$, Green: $(\frac{1}{20}, \frac{1}{5}, \frac{3}{4})$. More evenly distributed weights result in faster convergence.

**Initialization geometry.** We empirically study the bad initialization point $\boldsymbol{\mu}(0)$ described in Theorem 7 [2] by plotting the gradient norm at $\boldsymbol{\mu}(0)$ w.r.t. different dimension $d$ in Figure 2 (left). As theoretically analyzed, the gradient norm $\|\nabla \mathcal{L}(\boldsymbol{\mu}(0))\|$ at $\boldsymbol{\mu}(0)$ decreases exponentially in dimension $d$.

**Statistical rates.** The statistical rate for EM/gradient EM is another interesting research problem, which we observe empirically in Figure 2 (right). We run gradient EM on 5-component GMM with equal weights. x-axis: number of training samples, y-axis: parametric error after convergence. For each sample size, we run 50 times and report the average. The statistical errors are reported in the blue line. The red line (function $\Theta(n^{-1/4})$) and green line (linear regression output fitting blue points) are references. The trajectory approximately follows the law of $accuracy \propto n^{-1/4}$. While [Wu and Zhou, 2019] rigorously proves the asymptotic statistical rate of $\tilde{O}(n^{-1/4})$ for the special

---

[2]To prevent numerical underflow issues, we change the constant 12 in $\boldsymbol{\mu}(0)$ to 2.

case of 2-GMMs, our experiments imply that the same rate might also apply to the general case of multi-component GMMs.

## 6 Conclusion

This paper gives the first global convergence of gradient EM for over-parameterized Gaussian mixture models when the ground truth is a single Gaussian, and rate is sublinear which is exponentially slower than the rate in the exact-parameterization case. One fundamental open problem is to study when one can obtain global convergence of EM or gradient EM for Gaussian mixture models when the ground truth has multiple components. The likelihood-based convergence framework proposed in this paper might be an helpful tool towards solving this general problem.

**Acknowledgements** This work was supported in part by the following grants: NSF TRIPODS II-DMS 20231660, NSF CCF 2212261, NSF CCF 2007036, NSF AF 2312775, NSF IIS 2110170, NSF DMS 2134106, NSF IIS 2143493, and NSF IIS 2229881.

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

# A  Missing Proofs and Auxiliary lemmas

*Proof of Fact 1.* It is well known that (see Section 1 of Wu and Zhou [2019])

$$Q(\boldsymbol{\mu}'|\boldsymbol{\mu}) = \mathbf{E}_{x \sim p_{\boldsymbol{\mu}^*}} \left[ \log(p_{\boldsymbol{\mu}'}(x)) - D_{\mathrm{KL}}(p_{\boldsymbol{\mu}}(\cdot|x) || p_{\boldsymbol{\mu}'}(\cdot|x)) - H(p_{\boldsymbol{\mu}}(\cdot|x)) \right],$$

where $p_{\boldsymbol{\mu}}(\cdot|x)$ denotes the distribution of hidden variable $y$ (in our case of GMM the index of Gaussian component) conditioned on $x$, and $H$ denotes information entropy.

Since $\boldsymbol{\mu}' = \boldsymbol{\mu}$ is a global minimum of $D_{\mathrm{KL}}(p_{\boldsymbol{\mu}}(\cdot|x) || p_{\boldsymbol{\mu}'}(\cdot|x))$, we have $\nabla D_{\mathrm{KL}}(p_{\boldsymbol{\mu}}(\cdot|x) || p_{\boldsymbol{\mu}'}(\cdot|x)) = 0$. Also $\nabla H(p_{\boldsymbol{\mu}}(\cdot|x)) = 0$ since $H(p_{\boldsymbol{\mu}}(\cdot|x))$ is a constant. Therefore

$$\nabla Q(\boldsymbol{\mu}|\boldsymbol{\mu}) = \mathbf{E}_{x \sim p_{\boldsymbol{\mu}^*}} \left[ \nabla \log(p_{\boldsymbol{\mu}}(x)) \right] = \nabla \mathcal{L}(\boldsymbol{\mu}).$$

$\square$

The proof of Lemma 9 uses ideas from Theorem 1 of Chen et al. [2023] and relies on Stein's identity, which is given by the following lemma.

**Lemma 16** (Stein [1981]). *For $x \sim \mathcal{N}(\mu, \sigma^2 I_d)$ and differentiable function $g : \mathbf{R}^d \to \mathbf{R}$ we have*

$$\mathbf{E}[g(x)(x - \mu)] = \sigma^2 \mathbf{E}[\nabla_x g(x)],$$

*if the two expectations in the above identity exist.*

Now we are ready to prove Lemma 9.

**Lemma 9.** *For any GMM($\boldsymbol{\mu}$), $i \in [n]$, the gradient of Q satisfies*

$$\nabla_{\mu_i} \mathcal{L}(\boldsymbol{\mu}) = \nabla_{\mu_i} Q(\boldsymbol{\mu}|\boldsymbol{\mu}) = \mathbf{E}_x \left[ \psi_i(x) \sum_{k \in [n]} \psi_k(x) \mu_k \right].$$

*Proof.* Applying Stein's identity (Lemma 16), for each $i \in [n]$ we have

$$\begin{aligned}
\nabla_{\mu_i} Q(\boldsymbol{\mu}|\boldsymbol{\mu}) &= \mathbf{E}_{x \sim \mathcal{N}(0, I_d)} \left[ \psi_i(x)(\mu_i - x) \right] \\
&= \mathbf{E}_{x \sim \mathcal{N}(0, I_d)} \left[ \psi_i(x) \right] \mu_i - \mathbf{E}_{x \sim \mathcal{N}(0, I_d)} \left[ \psi_i(x) x \right] \\
&= \mathbf{E}_{x \sim \mathcal{N}(0, I_d)} \left[ \psi_i(x) \right] \mu_i - \mathbf{E}_{x \sim \mathcal{N}(0, I_d)} [\nabla_x \psi_i(x)].
\end{aligned}$$

Recall that

$$\psi_i(x) = \Pr[i|x] = \frac{\pi_i \exp\left(-\frac{\|x - \mu_i\|^2}{2}\right)}{\sum_{k \in [n]} \pi_k \exp\left(-\frac{\|x - \mu_k\|^2}{2}\right)}.$$

The gradient $\nabla_x \psi_i(x)$ could be calculated as

$$\begin{aligned}
&\nabla_x \psi_i(x) \\
&= \frac{1}{\left(\sum_{k \in [n]} \pi_k \exp\left(-\frac{\|x - \mu_k\|^2}{2}\right)\right)^2} \left[ \left( \sum_{k \in [n]} \pi_k \exp\left(-\frac{\|x - \mu_k\|^2}{2}\right) \right) \pi_i \exp\left(-\frac{\|x - \mu_i\|^2}{2}\right) (\mu_i - x) \right. \\
&\qquad \left. - \pi_i \exp\left(-\frac{\|x - \mu_i\|^2}{2}\right) \left( \sum_{k \in [n]} \pi_k \exp\left(-\frac{\|x - \mu_k\|^2}{2}\right) (\mu_k - x) \right) \right] \\
&= \psi_i(x)(\mu_i - x) - \psi_i(x) \sum_{k \in [n]} \psi_k(x)(\mu_k - x) \\
&= \psi_i(x)(\mu_i - x) + \psi_i(x) x - \sum_{k \in [n]} \psi_i(x) \psi_k(x) \mu_k \\
&= \psi_i(x) \left( \mu_i - \sum_{k \in [n]} \psi_k(x) \mu_k \right),
\end{aligned}$$

$$\tag{8}$$

note that we used $\sum_{k \in [n]} \psi_i(x) = 1$.

Then we have

$$\nabla_{\mu_i} Q(\boldsymbol{\mu}|\boldsymbol{\mu}) = \mathbf{E}_x \left[ \psi_i(x) \right] \mu_i - \mathbf{E}_x [\nabla_x \psi_i(x)]$$

$$= \mathbf{E}_x \left[ \psi_i(x) \right] \mu_i - \mathbf{E}_x \left[ \psi_i(x) \left( \mu_i - \sum_{k \in [n]} \psi_k(x)\mu_k \right) \right] = \mathbf{E}_x \left[ \psi_i(x) \sum_{k \in [n]} \psi_k(x)\mu_k \right].$$

$\square$

*Proof of Corollary 10.*

$$\langle \nabla_{\boldsymbol{\mu}} Q(\boldsymbol{\mu}|\boldsymbol{\mu}), \boldsymbol{\mu} \rangle = \sum_{i \in [n]} \langle \nabla_{\mu_i} Q(\boldsymbol{\mu}|\boldsymbol{\mu}), \mu_i \rangle = \sum_{i \in [n]} \left\langle \mathbf{E}_x \left[ \psi_i(x) \sum_{k \in [n]} \psi_k(x)\mu_k \right], \mu_i \right\rangle$$

$$= \sum_{i \in [n]} \sum_{k \in [n]} \mathbf{E}_x \langle \psi_i(x)\psi_k(x)\mu_k, \mu_i \rangle = \mathbf{E}_x \left[ \left\| \sum_{i \in [n]} \psi_i(x)\mu_i \right\|^2 \right] = \mathbf{E}_x \left[ \left\| \tilde{\psi}_{\boldsymbol{\mu}}(x) \right\|^2 \right].$$

$\square$

**Lemma 17.** *For any constant $c$ satisfying $0 < c \leq \frac{1}{3d}$, we have*

$$\mathbf{E}_{x \sim \mathcal{N}(0, I_d)} \left[ \exp\left( c\|x\| \right) \right] \leq 1 + 5\sqrt{d}c.$$

*Proof.* Note that $\mathbf{E}_{x \sim \mathcal{N}(0, I_d)} \left[ \exp\left( c\|x\| \right) \right] = \mathcal{M}_{\|x\|}(c)$ is the moment-generating function of $\|x\|$. To upper bound the value of a moment generating function at $c$, we use Lagrange's Mean Value Theorem:

$$\mathcal{M}_{\|x\|}(c) = \mathcal{M}_{\|x\|}(0) + \mathcal{M}'_{\|x\|}(\xi)c, \tag{9}$$

where $\xi \in [0, c]$. Note that $\mathcal{M}_{\|x\|}(0) = 1$, So the remaining task is to bound $\mathcal{M}'_{\|x\|}(\xi)$. We bound this expectation using truncation method as:

$$\mathcal{M}'_{\|x\|}(\xi) = \mathbf{E}_x \left[ \|x\| \exp(\xi\|x\|) \right] \leq \mathbf{E}_x \left[ \|x\| \exp(c\|x\|) \right]$$

$$= \int_{x \in \mathbf{R}^d} \|x\| \exp(c\|x\|)(2\pi)^{-d/2} \exp\left( -\frac{\|x\|^2}{2} \right) \mathrm{d}x$$

$$= \int_{\|x\| \leq 1} \|x\| \exp(c\|x\|)(2\pi)^{-d/2} \exp\left( -\frac{\|x\|^2}{2} \right) \mathrm{d}x$$

$$+ \int_{\|x\| \geq 1} \|x\| \exp(c\|x\|)(2\pi)^{-d/2} \exp\left( -\frac{\|x\|^2}{2} \right) \mathrm{d}x \tag{10}$$

$$\leq \exp(c)(2\pi)^{-d/2} V_d + \int_{\|x\| \geq 1} \|x\|(2\pi)^{-d/2} \exp\left( c\|x\| - \frac{\|x\|^2}{2} \right) \mathrm{d}x$$

$$\leq \exp(c)(2\pi)^{-d/2} V_d + \int_{\|x\| \geq 1} \|x\|(2\pi)^{-d/2} \exp\left( c\|x\| - \frac{\|x\|^2}{2} \right) \mathrm{d}x,$$

where $V_d = \frac{\pi^{d/2}}{\Gamma(d/2+1)}$ is the volume of $d$-dimensional unit sphere.

Since $\|x\| \geq 1 \Rightarrow c\|x\| - \frac{\|x\|^2}{2} \leq \frac{1}{3d}\|x\| - \frac{\|x\|^2}{2} \leq -\frac{\|(1-1/(2d))x\|^2}{2}$, we have

$$
\begin{aligned}
\int_{\|x\| \geq 1} & \|x\|(2\pi)^{-d/2} \exp\left(c\|x\| - \frac{\|x\|^2}{2}\right) \mathrm{d}x \\
&\leq \int_{\|x\| \geq 1} \|x\|(2\pi)^{-d/2} \exp\left(-\frac{\|\frac{2d-1}{2d}x\|^2}{2}\right) \mathrm{d}x \\
&= \int_{\|y\| \geq \frac{2d-1}{2d}} \frac{2d}{2d-1}\|y\|(2\pi)^{-d/2} \exp\left(-\frac{\|y\|^2}{2}\right) \left(\frac{2d}{2d-1}\right)^d \mathrm{d}y \\
&\leq \left(\frac{2d}{2d-1}\right)^{d+1} \mathbf{E}_{y \sim \mathcal{N}(0, I_d)} [\|y\|] \\
&= \left(\frac{2d}{2d-1}\right)^{d+1} \frac{\sqrt{2}\Gamma\left(\frac{d+1}{2}\right)}{\Gamma\left(\frac{d}{2}\right)} \\
&\leq 4\sqrt{d},
\end{aligned}
$$

where we used $\left(\frac{2d}{2d-1}\right)^{d+1} \leq 4$ and the log convexity of Gamma function at the last line. Plugging this back to (10), we get

$$
\begin{aligned}
\mathcal{M}'_{\|x\|}(\xi) &\leq \exp(c)(2\pi)^{-d/2}V_d + \int_{\|x\| \geq 1} \|x\|(2\pi)^{-d/2} \exp\left(c\|x\| - \frac{\|x\|^2}{2}\right) \mathrm{d}x \\
&\leq \exp(1/(3d))(2\pi)^{-d/2} + 4\sqrt{d} \\
&\leq 5\sqrt{d}.
\end{aligned}
\tag{11}
$$

Plugging (11) into (9), we obtain the final bound

$$
\mathbf{E}_x\left[\exp\left(2\|\delta_i\|(\|x\| + \|\mu_i\|)\right) - 1\right] = \mathcal{M}_{\|x\|}(c) = \mathcal{M}_{\|x\|}(0) + \mathcal{M}'_{\|x\|}(\xi)c \leq 1 + 5\sqrt{d}c.
$$

$\square$

**Lemma 18.** *Recall that $U = \sum_{i \in [n]} \|\mu_i\|^2$. For any fixed $x \in \mathbf{R}^d$, $x \neq 0$ and any $\boldsymbol{\mu}$ we have*

$$
\int_{t=-1}^1 \psi_i(tx|\boldsymbol{\mu})\psi_j(tx|\boldsymbol{\mu})\mathrm{d}t \geq \frac{1}{2\mu_{\max}\|x\|}\pi_i\pi_j \exp(-4U)(1 - \exp(-4\mu_{\max}\|x\|)).
$$

*Proof.*

$$
\begin{aligned}
\psi_i(tx) &= \frac{\pi_i \exp\left(-\frac{\|tx - \mu_i\|^2}{2}\right)}{\sum_{k \in [n]} \pi_k \exp\left(-\frac{\|tx - \mu_k\|^2}{2}\right)} \\
&= \frac{\pi_i}{\sum_{k \in [n]} \pi_k \exp\left(\frac{1}{2}(\|tx - \mu_i\|^2 - \|tx - \mu_k\|^2)\right)} \\
&= \frac{\pi_i}{\sum_{k \in [n]} \pi_k \exp\left(\frac{1}{2}(\|tx - \mu_i\|^2 - \|tx - \mu_k\|^2)\right)} \\
&= \frac{\pi_i}{\sum_{k \in [n]} \pi_k \exp\left(\frac{1}{2}\langle 2tx - \mu_i - \mu_k, \mu_k - \mu_i\rangle\right)} \\
&\geq \frac{\pi_i}{\sum_{k \in [n]} \pi_k \exp\left(\frac{1}{2}(2\|tx\| + 2\mu_{\max}) \cdot 2\mu_{\max}\right)} \\
&= \pi_i \exp(-2\mu_{\max}(\|tx\| + \mu_{\max}))
\end{aligned}
\tag{12}
$$

Therefore

$$\int_{t=-1}^{1} \psi_i(tx)\psi_j(tx)\mathrm{d}t \geq \int_{t=-1}^{1} \pi_i\pi_j \exp\left(-4\mu_{\max}(\|tx\| + \mu_{\max})\right)\mathrm{d}t$$

$$= \pi_i\pi_j \exp\left(-4\mu_{\max}^2\right) \cdot 2\int_{t=0}^{1} \exp\left(-4\mu_{\max}\|x\|t\right)\mathrm{d}t \qquad (13)$$

$$\geq \frac{1}{2\mu_{\max}\|x\|}\pi_i\pi_j \exp\left(-4U\right)\left(1 - \exp\left(-4\mu_{\max}\|x\|\right)\right).$$

$\square$

# B  Proofs for Section 3 and 4

## B.1  Proofs for global convergence analysis

**Theorem 13.** *At any two points* $\boldsymbol{\mu} = (\mu_1^\top, \ldots, \mu_n^\top)^\top$ *and* $\boldsymbol{\mu} + \boldsymbol{\delta} = ((\mu_1 + \delta_1)^\top, \ldots, (\mu_n + \delta_n)^\top)^\top$, *if*

$$\|\delta_i\| \leq \frac{1}{\max\{6d, 2\|\mu_i\|\}}, \forall i \in [n],$$

*then the loss function* $\mathcal{L}$ *satisfies the following smoothness property: for any* $i \in [n]$ *we have*

$$\|\nabla_{\mu_i+\delta_i}\mathcal{L}(\boldsymbol{\mu} + \boldsymbol{\delta}) - \nabla_{\mu_i}\mathcal{L}(\boldsymbol{\mu})\| \leq n\mu_{\max}(30\sqrt{d} + 4\mu_{\max})\|\delta_i\| + \sum_{k\in[n]}\|\delta_k\|. \qquad (14)$$

*Proof.* Note that

$$\exp\left(-\|\delta_i\|(\|x\| + \|\mu_i\|)\right)\exp\left(-\frac{\|\delta_i\|^2}{2}\right) \leq \frac{\exp\left(-\frac{\|x-(\mu_i+\delta_i)\|^2}{2}\right)}{\exp\left(-\frac{\|x-\mu_i\|^2}{2}\right)} = \exp\left(\langle x - \mu_i, \delta_i\rangle - \frac{\|\delta_i\|^2}{2}\right)$$

$$\leq \exp\left(\|\delta_i\|(\|x\| + \|\mu_i\|)\right)\exp\left(-\frac{\|\delta_i\|^2}{2}\right).$$

Therefore $\psi_i(x|\boldsymbol{\mu} + \boldsymbol{\delta})$ can be bounded as

$$\psi_i(x|\boldsymbol{\mu} + \boldsymbol{\delta}) = \frac{\pi_i \exp\left(-\frac{\|x-(\mu_i+\delta_i)\|^2}{2}\right)}{\sum_{k\in[n]} \pi_k \exp\left(-\frac{\|x-(\mu_k+\delta_k)\|^2}{2}\right)}$$

$$\leq \frac{\pi_i \exp\left(-\frac{\|x-\mu_i\|^2}{2}\right)\exp\left(\|\delta_i\|(\|x\| + \|\mu_i\|)\right)\exp\left(-\frac{\|\delta_i\|^2}{2}\right)}{\sum_{k\in[n]} \pi_k \exp\left(-\frac{\|x-\mu_k\|^2}{2}\right)\exp\left(-\|\delta_i\|(\|x\| + \|\mu_i\|)\right)\exp\left(-\frac{\|\delta_i\|^2}{2}\right)} \leq \exp\left(2\|\delta_i\|(\|x\| + \|\mu_i\|)\right)\psi_i(x|\boldsymbol{\mu}).$$

$$(15)$$

Similarly, we have

$$\psi_i(x|\boldsymbol{\mu} + \boldsymbol{\delta}) = \frac{\pi_i \exp\left(-\frac{\|x-(\mu_i+\delta_i)\|^2}{2}\right)}{\sum_{k\in[n]} \pi_k \exp\left(-\frac{\|x-(\mu_k+\delta_k)\|^2}{2}\right)}$$

$$\geq \frac{\pi_i \exp\left(-\frac{\|x-\mu_i\|^2}{2}\right)\exp\left(-\|\delta_i\|(\|x\| + \|\mu_i\|)\right)\exp\left(-\frac{\|\delta_i\|^2}{2}\right)}{\sum_{k\in[n]} \pi_k \exp\left(-\frac{\|x-\mu_k\|^2}{2}\right)\exp\left(\|\delta_i\|(\|x\| + \|\mu_i\|)\right)\exp\left(-\frac{\|\delta_i\|^2}{2}\right)} \geq \exp\left(-2\|\delta_i\|(\|x\| + \|\mu_i\|)\right)\psi_i(x|\boldsymbol{\mu}).$$

$$(16)$$

Recall that by Lemma 9 we have $\nabla_{\mu_i}\mathcal{L}(\boldsymbol{\mu}) = \mathbf{E}_x\left[\psi_i(x|\boldsymbol{\mu})\sum_{k\in[n]}\psi_k(x|\boldsymbol{\mu})\mu_k\right]$, so

$$
\begin{aligned}
&\|\nabla_{\mu_i+\delta_i}\mathcal{L}(\boldsymbol{\mu}+\boldsymbol{\delta}) - \nabla_{\mu_i}\mathcal{L}(\boldsymbol{\mu})\| \\
&= \left\|\mathbf{E}_x\left[\psi_i(x|\boldsymbol{\mu}+\boldsymbol{\delta})\sum_{k\in[n]}\psi_k(x|\boldsymbol{\mu}+\boldsymbol{\delta})(\mu_k+\delta_k)\right] - \mathbf{E}_x\left[\psi_i(x|\boldsymbol{\mu})\sum_{k\in[n]}\psi_k(x|\boldsymbol{\mu})\mu_k\right]\right\| \\
&= \left\|\mathbf{E}_x\left[\sum_{k\in[n]}\psi_i(x|\boldsymbol{\mu}+\boldsymbol{\delta})\psi_k(x|\boldsymbol{\mu}+\boldsymbol{\delta})\delta_k\right]\right. \\
&\quad + \left.\mathbf{E}_x\left[\sum_{k\in[n]}(\psi_i(x|\boldsymbol{\mu}+\boldsymbol{\delta})\psi_k(x|\boldsymbol{\mu}+\boldsymbol{\delta}) - \psi_i(x|\boldsymbol{\mu})\psi_k(x|\boldsymbol{\mu}))\mu_k\right]\right\| \\
&\leq \mathbf{E}_x\left[\sum_{k\in[n]}\psi_i(x|\boldsymbol{\mu}+\boldsymbol{\delta})\psi_k(x|\boldsymbol{\mu}+\boldsymbol{\delta})\|\delta_k\|\right] \\
&\quad + \mathbf{E}_x\left[\sum_{k\in[n]}|\psi_i(x|\boldsymbol{\mu}+\boldsymbol{\delta})\psi_k(x|\boldsymbol{\mu}+\boldsymbol{\delta}) - \psi_i(x|\boldsymbol{\mu})\psi_k(x|\boldsymbol{\mu})|\cdot\|\mu_k\|\right] \\
&\leq \sum_{k\in[n]}\|\delta_k\| + \sum_{k\in[n]}\mathbf{E}_x\left[|\psi_i(x|\boldsymbol{\mu}+\boldsymbol{\delta})\psi_k(x|\boldsymbol{\mu}+\boldsymbol{\delta}) - \psi_i(x|\boldsymbol{\mu})\psi_k(x|\boldsymbol{\mu})|\right]\|\mu_k\| \\
&\leq \sum_{k\in[n]}\|\delta_k\| + \sum_{k\in[n]}\mathbf{E}_x\left[\exp\left(2\|\delta_i\|(\|x\|+\|\mu_i\|)\right)-1\right]\|\mu_k\|,
\end{aligned}
\tag{17}
$$

where the last inequality is because $\psi_i, \psi_k \leq 1$ and applying (15) and (16).

The remaining task is to bound $\mathbf{E}_x\left[\exp\left(2\|\delta_i\|(\|x\|+\|\mu_i\|)\right)-1\right]$. Since $2\|\delta_i\| \leq \frac{1}{3d}$, we can use Lemma 17 to bound it as

$$
\begin{aligned}
&\mathbf{E}_x\left[\exp\left(2\|\delta_i\|(\|x\|+\|\mu_i\|)\right)-1\right] = \exp(2\|\delta_i\|\|\mu_i\|)\mathbf{E}_x\left[\exp\left(2\|\delta_i\|\cdot\|x\|)\right)\right]-1 \\
&\leq \exp(2\|\delta_i\|\|\mu_i\|)(1+10\sqrt{d}\|\delta_i\|)-1 = \exp(2\|\delta_i\|\|\mu_i\|)-1+10\sqrt{d}\|\delta_i\|\exp(2\|\delta_i\|\|\mu_i\|) \\
&\leq 4\|\delta_i\|\|\mu_i\| + 10\sqrt{d}\|\delta_i\|\exp(1) \leq (30\sqrt{d}+4\|\mu_i\|)\|\delta_i\|.
\end{aligned}
\tag{18}
$$

where we used $\exp(1+x) \leq 1+2x, \forall x \in [0,1]$ at the last line. Plugging this back to (17), we get

$$
\begin{aligned}
&\|\nabla_{\mu_i+\delta_i}\mathcal{L}(\boldsymbol{\mu}+\boldsymbol{\delta}) - \nabla_{\mu_i}\mathcal{L}(\boldsymbol{\mu})\| \\
&\leq \sum_{k\in[n]}\|\delta_k\| + \sum_{k\in[n]}\mathbf{E}_x\left[\exp\left(2\|\delta_i\|(\|x\|+\|\mu_i\|)\right)-1\right]\|\mu_k\| \\
&\leq \sum_{k\in[n]}\|\delta_k\| + \sum_{k\in[n]}(30\sqrt{d}+4\|\mu_i\|)\|\delta_i\|\|\mu_k\| \\
&\leq n\mu_{\max}(30\sqrt{d}+4\mu_{\max})\|\delta_i\| + \sum_{k\in[n]}\|\delta_k\|.
\end{aligned}
\tag{19}
$$

$\square$

**Theorem 14.** *The loss function can be upper bounded as*

$$
\mathcal{L}(\boldsymbol{\mu}) \leq \sum_{i\in[n]}\frac{\pi_i}{2}\|\mu_i\|^2 \leq \frac{\mu_{\max}^2}{2}.
$$

*Proof.* Since the logarithm function is concave, by Jensen's inequality we have

$$\mathcal{L}(\boldsymbol{\mu}) = D_{\mathrm{KL}}(p_{\boldsymbol{\mu}^*}\|p_{\boldsymbol{\mu}}) = -\mathbf{E}_x\left[\log\left(\frac{p_{\boldsymbol{\mu}}(x)}{p_{\boldsymbol{\mu}^*}(x)}\right)\right]$$

$$= -\mathbf{E}_x\left[\log\left(\frac{\sum_i \pi_i \exp\left(-\frac{\|x-\mu_i\|^2}{2}\right)}{\exp\left(-\frac{\|x\|^2}{2}\right)}\right)\right]$$

$$\leq -\mathbf{E}_x\left[\sum_i \pi_i \log\left(\frac{\exp\left(-\frac{\|x-\mu_i\|^2}{2}\right)}{\exp\left(-\frac{\|x\|^2}{2}\right)}\right)\right]$$

$$= -\sum_i \pi_i \mathbf{E}_x\left[\langle x, \mu_i\rangle - \frac{\|\mu_i\|^2}{2}\right]$$

$$= \sum_{i\in[n]} \frac{\pi_i}{2}\|\mu_i\|^2 \leq \frac{\mu_{\max}^2}{2}.$$

$\square$

**Lemma 12.** *For any GMM$(\boldsymbol{\mu})$ we have*

$$\langle \nabla_{\boldsymbol{\mu}} Q(\boldsymbol{\mu}|\boldsymbol{\mu}), \boldsymbol{\mu}\rangle = \mathbf{E}_x[\|\tilde{\boldsymbol{\psi}}_{\boldsymbol{\mu}}(x)\|^2] \geq \Omega\left(\frac{\exp(-8U)\,\pi_{\min}^2}{d(1+\mu_{\max}\sqrt{d})^2}\mu_{\max}^4\right).$$

*Proof.* Consider two cases:

**Case 1.** There exists $k \in [n]$ such that $\|\mu_k - \mu_{i_{\max}}\| \geq \frac{\mu_{\max}}{2}$. Then by Lemma 19 and Lemma 11 we have

$$\mathbf{E}_x\left[\|\tilde{\boldsymbol{\psi}}_{\boldsymbol{\mu}}(x)\|^2\right] \geq \frac{\exp(-8U)}{40000d(1+2\mu_{\max}\sqrt{d})^2}\left(\sum_{i,j\in[n]}\pi_i\pi_j\|\mu_i-\mu_j\|^2\right)^2$$

$$\geq \frac{\exp(-8U)}{40000d(1+2\mu_{\max}\sqrt{d})^2}\left(\frac{\pi_{\min}}{8}\mu_{\max}^2\right)^2$$

$$= \frac{\exp(-8U)\,\pi_{\min}^2}{2560000d(1+2\mu_{\max}\sqrt{d})^2}\mu_{\max}^4.$$

**Case 2.** For $\forall k \in [n]$, $\|\mu_{i_{\max}} - \mu_k\| < \frac{\mu_{\max}}{2}$. Then by Lemma 20 we have $\mathbf{E}_x\left[\|\tilde{\boldsymbol{\psi}}_{\boldsymbol{\mu}}(x)\|^2\right] \geq \frac{1}{4}\mu_{\max}^2 \geq \Omega(\exp(-8\mu_{\max}^2)\mu_{\max}^4) \geq \Omega(\exp(-8U)\mu_{\max}^4) \geq \Omega\left(\frac{\exp(-8U)\pi_{\min}^2}{d(1+\mu_{\max}\sqrt{d})^2}\mu_{\max}^4\right)$, (since $e^{-x}x \leq 1, \forall x$). $\square$

**Lemma 19.** *For any GMM$(\boldsymbol{\mu})$, if there exists $k \in [n]$ such that $\|\mu_k - \mu_{i_{\max}}\| \geq \frac{\mu_{\max}}{2}$, then we have*

$$\sum_{i,j\in[n]}\pi_i\pi_j\|\mu_i-\mu_j\|^2 \geq \frac{\pi_{\min}}{8}\mu_{\max}^2.$$

*Proof.* By Cauchy–Schwarz inequality, we have $\|a\|^2 + \|b\|^2 \geq \frac{1}{2}\|a-b\|^2$, so for $\forall i \in [n]$ we have

$$\sum_{j\in[n]}\pi_j\|\mu_i-\mu_j\|^2 \geq \pi_{i_{\max}}\|\mu_i-\mu_{i_{\max}}\|^2 + \pi_k\|\mu_i-\mu_k\|^2$$

$$\geq \frac{\pi_{\min}}{2}\|(\mu_i-\mu_{i_{\max}})-(\mu_i-\mu_k)\|^2 = \frac{\pi_{\min}}{2}\|\mu_k-\mu_{i_{\max}}\|^2.$$

Therefore

$$\sum_{i,j\in[n]}\pi_i\pi_j\|\mu_i-\mu_j\|^2 = \sum_{i\in[n]}\pi_i\sum_{j\in[n]}\pi_j\|\mu_i-\mu_j\|^2 \geq \sum_{i\in[n]}\pi_i\frac{\pi_{\min}}{2}\|\mu_k-\mu_{i_{\max}}\|^2 \geq \frac{\pi_{\min}}{8}\mu_{\max}^2,$$

where the last inequality is because $\|\mu_k-\mu_{i_{\max}}\| \geq \frac{\mu_{\max}}{2}$ and $\sum_i \pi_i = 1$. $\square$

**Lemma 20.** *For any GMM($\boldsymbol{\mu}$), if for $\forall k \in [n]$ we have $\|\mu_{i_{\max}} - \mu_k\| < \frac{\mu_{\max}}{2}$, then*

$$\mathbf{E}_x \left[ \|\tilde{\psi}_{\boldsymbol{\mu}}(x)\|^2 \right] \geq \frac{1}{4} \mu_{\max}^2.$$

*Proof.* For any $k \in [n]$, by Cauchy–Schwarz inequality we have

$$\langle \mu_k, \mu_{i_{\max}} \rangle = \langle \mu_{i_{\max}} - (\mu_{i_{\max}} - \mu_k), \mu_{i_{\max}} \rangle = \|\mu_{i_{\max}}\|^2 - \langle \mu_{i_{\max}} - \mu_k, \mu_{i_{\max}} \rangle$$
$$\geq \mu_{\max}^2 - \|\mu_{i_{\max}} - \mu_k\| \mu_{\max} > \frac{1}{2} \mu_{\max}^2, \tag{20}$$

where the last inequality is because $\|\mu_{i_{\max}} - \mu_k\| < \frac{\mu_{\max}}{2}$.

Note that (20) implies $\langle \mu_k, \overline{\mu_{i_{\max}}} \rangle > \frac{1}{2}\mu_{\max}$, so for $\forall x \in \mathbf{R}^d$ we have

$$\|\tilde{\psi}_{\boldsymbol{\mu}}(x)\| = \left\| \sum_{k \in [n]} \psi_k(x)\mu_k \right\| \geq \left\langle \sum_{k \in [n]} \psi_k(x)\mu_k, \overline{\mu_{i_{\max}}} \right\rangle = \sum_{k \in [n]} \psi_k(x) \langle \mu_k, \overline{\mu_{i_{\max}}} \rangle > \frac{1}{2}\mu_{\max},$$
$$\tag{21}$$

where we used $\sum_{k \in [n]} \psi_k(x) = 1$ at the last inequality. $\square$

**Lemma 11.** *For any GMM($\boldsymbol{\mu}$) we have*

$$\mathbf{E}_x \left[ \|\tilde{\psi}_{\boldsymbol{\mu}}(x)\|^2 \right] \geq \frac{\exp(-8U)}{40000d(1 + 2\mu_{\max}\sqrt{d})^2} \left( \sum_{i,j \in [n]} \pi_i \pi_j \|\mu_i - \mu_j\|^2 \right)^2.$$

*Proof.* The key idea is to consider the gradient of $\tilde{\psi}_{\boldsymbol{\mu}}$, which can be calculated as

$$\nabla_x \tilde{\psi}_{\boldsymbol{\mu}}(x) = \sum_i \mu_i \left( \frac{\partial \psi_i(x)}{\partial x} \right)^\top$$
$$= \sum_i \psi_i(x)\mu_i\mu_i^\top - \sum_{i,j} \psi_i(x)\psi_j(x)\mu_i\mu_j^\top$$
$$= \sum_{i,j \in [n]} \psi_i(x)\psi_j(x)\mu_i\mu_i^\top - \sum_{i,j} \psi_i(x)\psi_j(x)\mu_i\mu_j^\top$$
$$= \sum_{i,j \in [n]} \psi_i(x)\psi_j(x)\mu_i(\mu_i - \mu_j)^\top \tag{22}$$
$$= \sum_{i,j \in [n]} \psi_i(x)\psi_j(x)\frac{1}{2} \left( \mu_i(\mu_i - \mu_j)^\top + \mu_j(\mu_j - \mu_i)^\top \right)$$
$$= \frac{1}{2} \sum_{i,j \in [n]} \psi_i(x)\psi_j(x)(\mu_i - \mu_j)(\mu_i - \mu_j)^\top,$$

where we used (8) in the second identity.

By Cauchy-Schwarz inequality, we have $\|a\|^2 + \|b\|^2 \geq \frac{1}{2}\|a-b\|^2$, which implies

$$
\begin{aligned}
\mathbf{E}_x\left[\|\tilde{\psi}_{\boldsymbol{\mu}}(x)\|^2\right] &= \frac{1}{2}\mathbf{E}_x\left[\|\tilde{\psi}_{\boldsymbol{\mu}}(x)\|^2 + \|\tilde{\psi}_{\boldsymbol{\mu}}(-x)\|^2\right] \\
&\geq \frac{1}{4}\mathbf{E}_x\left[\left\|\tilde{\psi}_{\boldsymbol{\mu}}(x) - \tilde{\psi}_{\boldsymbol{\mu}}(-x)\right\|^2\right] \\
&\geq \frac{1}{4}\mathbf{E}_x\left[\left\langle\tilde{\psi}_{\boldsymbol{\mu}}(x) - \tilde{\psi}_{\boldsymbol{\mu}}(-x), \overline{x}\right\rangle^2\right] \\
&= \frac{1}{4}\mathbf{E}_x\left[\left(\int_{t=-1}^1 \frac{\partial}{\partial t}\langle\tilde{\psi}_{\boldsymbol{\mu}}(tx), \overline{x}\rangle \mathrm{d}t\right)^2\right] \\
&= \frac{1}{4}\mathbf{E}_x\left[\left(\int_{t=-1}^1 x^\top \nabla\tilde{\psi}_{\boldsymbol{\mu}}(tx)\overline{x}\mathrm{d}t\right)^2\right] \\
&= \frac{1}{4}\mathbf{E}_x\left[\left(\int_{t=-1}^1 \|x\|\cdot\overline{x}^\top\nabla\tilde{\psi}_{\boldsymbol{\mu}}(tx)\overline{x}\mathrm{d}t\right)^2\right],
\end{aligned}
\tag{23}
$$

where we used $\frac{\partial}{\partial t}\tilde{\psi}_{\boldsymbol{\mu}}(tx) = \nabla\tilde{\psi}_{\boldsymbol{\mu}}(tx)x$ at the second to last identity. Careful readers might notice that the term $\left(\int_{t=-1}^1 \|x\|\cdot\overline{x}^\top\nabla\tilde{\psi}_{\boldsymbol{\mu}}(tx)\overline{x}\mathrm{d}t\right)^2$ is not well-defined when $x = 0$, but we can still calculate its expectation over the whole probability space since the integration is only singular on a zero-measure set.

For each $x \neq 0$, by (22) we have

$$
\overline{x}^\top\nabla\tilde{\psi}_{\boldsymbol{\mu}}(tx)\overline{x} = \frac{1}{2}\sum_{i,j\in[n]}\psi_i(tx)\psi_j(tx)\langle\mu_i - \mu_j, \overline{x}\rangle^2.
$$

So

$$
\begin{aligned}
&\mathbf{E}_x\left[\|\tilde{\psi}_{\boldsymbol{\mu}}(x)\|^2\right] \\
&\geq \frac{1}{16}\mathbf{E}_x\left[\left(\int_{t=-1}^1 \|x\|\sum_{i,j\in[n]}\psi_i(tx)\psi_j(tx)\langle\mu_i - \mu_j, \overline{x}\rangle^2\mathrm{d}t\right)^2\right] \\
&= \frac{1}{16}\mathbf{E}_x\left[\left(\|x\|\sum_{i,j\in[n]}\langle\mu_i - \mu_j, \overline{x}\rangle^2\int_{t=-1}^1 \psi_i(tx)\psi_j(tx)\mathrm{d}t\right)^2\right] \\
&\geq \frac{1}{16}\mathbf{E}_x\left[\left(\|x\|\sum_{i,j\in[n]}\langle\mu_i - \mu_j, \overline{x}\rangle^2\frac{1}{2\mu_{\max}\|x\|}\pi_i\pi_j\exp\left(-4U\right)\left(1 - \exp\left(-4\mu_{\max}\|x\|\right)\right)\right)^2\right] \\
&= \frac{\exp\left(-8U\right)}{64}\mathbf{E}_x\left[\left(\sum_{i,j\in[n]}\pi_i\pi_j\langle\mu_i - \mu_j, \overline{x}\rangle^2\frac{1 - \exp\left(-4\mu_{\max}\|x\|\right)}{\mu_{\max}}\right)^2\right] \\
&\geq \frac{\exp\left(-8U\right)}{64}\left(\sum_{i,j\in[n]}\pi_i\pi_j\mathbf{E}_x\left[\langle\mu_i - \mu_j, \overline{x}\rangle^2\frac{1 - \exp\left(-4\mu_{\max}\|x\|\right)}{\mu_{\max}}\right]\right)^2
\end{aligned}
\tag{24}
$$

where we used Lemma 18 at the fourth line and Cauchy-Schwarz inequality at the last line.

The last step is to lower bound $\mathbf{E}_x\left[\langle\mu_i - \mu_j, \overline{x}\rangle^2\left(1 - \exp\left(-4\mu_{\max}\|x\|\right)\right)/\mu_{\max}\right]$. Since $x$ is sampled from $\mathcal{N}(0, I_d)$, which is spherically symmetric, we know that the two random variables $\{\overline{x}, \|x\|\}$

are independent. Therefore

$$\mathbf{E}_x \left[ \langle \mu_i - \mu_j, \overline{x} \rangle^2 \frac{1 - \exp\left(-4\mu_{\max}\|x\|\right)}{\mu_{\max}} \right] = \mathbf{E}_x \left[ \langle \mu_i - \mu_j, \overline{x} \rangle^2 \right] \mathbf{E}_x \left[ \frac{1 - \exp\left(-4\mu_{\max}\|x\|\right)}{\mu_{\max}} \right]. \tag{25}$$

For the first term in (25), we have $\mathbf{E}_x \left[ \langle \mu_i - \mu_j, \overline{x} \rangle^2 \right] = \|\mu_i - \mu_j\|^2/d$ since $\overline{x}$ is spherically symmetrically distributed. By norm-concentration inequality of Gaussian [Dasgupta and Schulman, 2000] we know that $\Pr\left[ \|x\| \geq \frac{\sqrt{d}}{2} \right] \geq 1/50, \forall d$. The second term in (25) can be therefore lower bounded as

$$\mathbf{E}_x \left[ \frac{1 - \exp\left(-4\mu_{\max}\|x\|\right)}{\mu_{\max}} \right] \geq \Pr\left[ \|x\| \geq \frac{\sqrt{d}}{2} \right] \frac{1 - \exp\left(-4\mu_{\max} \cdot \frac{\sqrt{d}}{2}\right)}{\mu_{\max}} \geq \frac{1 - \exp\left(-2\mu_{\max}\sqrt{d}\right)}{50\mu_{\max}}. \tag{26}$$

Plugging (26) into (25), we get

$$\mathbf{E}_x \left[ \langle \mu_i - \mu_j, \overline{x} \rangle^2 \frac{1 - \exp\left(-4\mu_{\max}\|x\|\right)}{\mu_{\max}} \right] \geq \frac{1 - \exp\left(-2\mu_{\max}\sqrt{d}\right)}{50d\mu_{\max}} \|\mu_i - \mu_j\|^2. \tag{27}$$

Now we can plug (27) into (24) and get

$$
\begin{aligned}
\mathbf{E}_x \left[ \|\tilde{\psi}_{\boldsymbol{\mu}}(x)\|^2 \right] &\geq \frac{\exp\left(-8U\right)}{64} \left( \sum_{i,j\in[n]} \pi_i \pi_j \mathbf{E}_x \left[ \langle \mu_i - \mu_j, \overline{x} \rangle^2 \frac{1 - \exp\left(-4\mu_{\max}\|x\|\right)}{\mu_{\max}} \right] \right)^2 \\
&\geq \frac{\exp\left(-8U\right)}{64} \left( \sum_{i,j\in[n]} \pi_i \pi_j \frac{1 - \exp\left(-2\mu_{\max}\sqrt{d}\right)}{50d\mu_{\max}} \|\mu_i - \mu_j\|^2 \right)^2 \\
&\geq \frac{\exp\left(-8U\right)}{64} \left( \sum_{i,j\in[n]} \pi_i \pi_j \frac{1 - \frac{1}{1+2\mu_{\max}\sqrt{d}}}{50d\mu_{\max}} \|\mu_i - \mu_j\|^2 \right)^2 \\
&= \frac{\exp\left(-8U\right)}{40000d(1 + 2\mu_{\max}\sqrt{d})^2} \left( \sum_{i,j\in[n]} \pi_i \pi_j \|\mu_i - \mu_j\|^2 \right)^2
\end{aligned}
\tag{28}
$$

where we used the inequality $\forall t \geq 0, e^{-t} \leq \frac{1}{1+t}$ at the second to last line. $\qquad\square$

**Theorem 2.** *Consider training a student $n$-component GMM initialized from $\boldsymbol{\mu}(0) = (\mu_1(0)^\top, \ldots, \mu_n(0)^\top)^\top$ to learn a single-component ground truth GMM $\mathcal{N}(0, I_d)$ with population gradient EM algorithm. If the step size satisfies $\eta \leq O\left( \frac{\exp(-8U(0))\pi_{\min}^2}{n^2 d^2 (\frac{1}{\mu_{\max}(0)} + \mu_{\max}(0))^2} \right)$, then gradient EM converges globally with rate*

$$\mathcal{L}(\boldsymbol{\mu}(t)) \leq \frac{1}{\sqrt{\gamma t}},$$

*where $\gamma = \Omega\left( \frac{\eta \exp(-16U(0))\pi_{\min}^4}{n^2 d^2 (1+\mu_{\max}(0)\sqrt{dn})^4} \right) \in \mathbf{R}^+$. Recall that $\mu_{\max}(0) = \max\{\|\mu_1(0)\|, \ldots, \|\mu_n(0)\|\}$ and $U(0) = \sum_{i\in[n]} \|\mu_i(0)\|^2$ are two initialization constants.*

*Proof.* We use mathematical induction to prove Theorem 2, by proving the following two conditions inductively:

$$U(t) \leq U(0) = \sum_{i\in[n]} \|\mu_i(0)\|^2, \forall t. \tag{29}$$

$$\frac{1}{\mathcal{L}^2(\boldsymbol{\mu}(t))} \geq \gamma t + \frac{1}{\mathcal{L}^2(\boldsymbol{\mu}(0))}, \forall t. \tag{30}$$

Note that (30) directly implies the theorem, so now we just need to prove (29) and (30) together.

The induction base for $t = 0$ is trivial. Now suppose the conditions hold for time step $t$, consider $t + 1$. By induction hypothesis (29) we have $\|\mu_i(t)\| \leq \mu_{\max}(t) \leq \sqrt{n}\mu_{\max}(0), \forall t$.

**Proof of (30).** Since $\nabla_{\boldsymbol{\mu}}Q(\boldsymbol{\mu}|\boldsymbol{\mu}) = \nabla_{\boldsymbol{\mu}}\mathcal{L}(\boldsymbol{\mu})$, we can apply classical analysis of gradient descent [Nesterov et al., 2018] as

$$
\begin{aligned}
&\mathcal{L}(\boldsymbol{\mu}(t+1)) - \mathcal{L}(\boldsymbol{\mu}(t)) \\
&= \mathcal{L}(\boldsymbol{\mu}(t) - \eta\nabla\mathcal{L}(\boldsymbol{\mu}(t))) - \mathcal{L}(\boldsymbol{\mu}(t)) \\
&= -\int_{s=0}^{1} \langle \nabla\mathcal{L}(\boldsymbol{\mu}(t) - s\eta\nabla\mathcal{L}(\boldsymbol{\mu}(t))), \eta\nabla\mathcal{L}(\boldsymbol{\mu}(t)) \rangle \, \mathrm{d}s \\
&= -\int_{s=0}^{1} \langle \nabla\mathcal{L}(\boldsymbol{\mu}(t)), \eta\nabla\mathcal{L}(\boldsymbol{\mu}(t)) \rangle \, \mathrm{d}s + \int_{s=0}^{1} \langle \nabla\mathcal{L}(\boldsymbol{\mu}(t)) - \nabla\mathcal{L}(\boldsymbol{\mu}(t) - s\eta\nabla\mathcal{L}(\boldsymbol{\mu}(t))), \eta\nabla\mathcal{L}(\boldsymbol{\mu}(t)) \rangle \, \mathrm{d}s \\
&= -\eta\|\nabla\mathcal{L}(\boldsymbol{\mu}(t))\|^2 + \eta\int_{s=0}^{1} \langle \nabla\mathcal{L}(\boldsymbol{\mu}(t)) - \nabla\mathcal{L}(\boldsymbol{\mu}(t) - s\eta\nabla\mathcal{L}(\boldsymbol{\mu}(t))), \nabla\mathcal{L}(\boldsymbol{\mu}(t)) \rangle \, \mathrm{d}s
\end{aligned}
\tag{31}
$$

Note that the gradient norm can be upper bounded as

$$
\begin{aligned}
\|\nabla_{\mu_i}\mathcal{L}(\boldsymbol{\mu}(t))\| &= \left\| \mathbf{E}_x \left[ \psi_i(x) \sum_{k\in[n]} \psi_k(x)\mu_k(t) \right] \right\| \leq \mathbf{E}_x \left[ \psi_i(x) \sum_{k\in[n]} \psi_k(x) \|\mu_k(t)\| \right] \\
&\leq \sum_k \|\mu_k(t)\| \leq \sqrt{nU(t)} \leq n\mu_{\max}(0).
\end{aligned}
$$

Then for any $s \in [0,1]$, we have $\|s\eta\nabla_{\mu_i}\mathcal{L}(\boldsymbol{\mu}(t))\| \leq \eta n\mu_{\max}(0) \leq \frac{1}{\max\{6d,2\|\mu_i(t)\|\}}$. So we can apply Theorem 13 and get

$$
\begin{aligned}
&\|\nabla_{\mu_i}\mathcal{L}(\boldsymbol{\mu}(t)) - \nabla_{\mu_i}\mathcal{L}(\boldsymbol{\mu}(t) - s\eta\nabla_{\mu_i}\mathcal{L}(\boldsymbol{\mu}(t)))\| \\
&\leq n\mu_{\max}(t)(30\sqrt{d} + 4\mu_{\max}(t))\|s\eta\nabla_{\mu_i}\mathcal{L}(\boldsymbol{\mu}(t))\| + \sum_{k\in[n]} \|s\eta\nabla_{\mu_k}\mathcal{L}(\boldsymbol{\mu}(t))\|.
\end{aligned}
$$

Therefore for $\forall s \in [0,1]$,

$$
\begin{aligned}
&\langle \nabla\mathcal{L}(\boldsymbol{\mu}(t)) - \nabla\mathcal{L}(\boldsymbol{\mu}(t) - s\eta\nabla\mathcal{L}(\boldsymbol{\mu}(t))), \nabla\mathcal{L}(\boldsymbol{\mu}(t)) \rangle \\
&\leq \sum_{i\in[n]} \|\nabla_{\mu_i}\mathcal{L}(\boldsymbol{\mu}(t)) - \nabla_{\mu_i}\mathcal{L}(\boldsymbol{\mu}(t) - s\eta\nabla_{\mu_i}\mathcal{L}(\boldsymbol{\mu}(t)))\| \cdot \|\nabla_{\mu_i}\mathcal{L}(\boldsymbol{\mu}(t))\| \\
&\leq \sum_{i\in[n]} \left( n\mu_{\max}(t)(30\sqrt{d} + 4\mu_{\max}(t))\|s\eta\nabla_{\mu_i}\mathcal{L}(\boldsymbol{\mu}(t))\| + \sum_{k\in[n]} \|s\eta\nabla_{\mu_k}\mathcal{L}(\boldsymbol{\mu}(t))\| \right) \|\nabla_{\mu_i}\mathcal{L}(\boldsymbol{\mu}(t))\| \\
&\leq \eta \left( n\mu_{\max}(t)(30\sqrt{d} + 4\mu_{\max}(t)) + n^2 \right) \|\nabla\mathcal{L}(\boldsymbol{\mu}(t))\|^2 \\
&\leq \eta \left( 4n^2\mu_{\max}(0)^2 + 30\sqrt{d}n^{3/2}\mu_{\max}(0) + n^2 \right) \|\nabla\mathcal{L}(\boldsymbol{\mu}(t))\|^2 \\
&\leq 20\eta\sqrt{d}n^2(\mu_{\max}^2(0) + 1)\|\nabla\mathcal{L}(\boldsymbol{\mu}(t))\|^2.
\end{aligned}
\tag{32}
$$

Plugging (32) into (31), since $\eta \leq O\left( \frac{1}{\sqrt{d}n^2(\mu_{\max}^2(0)+1)} \right)$ we have

$$
\mathcal{L}(\boldsymbol{\mu}(t+1)) - \mathcal{L}(\boldsymbol{\mu}(t)) \leq -\eta\|\nabla\mathcal{L}(\boldsymbol{\mu}(t))\|^2 + 20\eta\sqrt{d}n^2(\mu_{\max}^2(0)+1)\|\nabla\mathcal{L}(\boldsymbol{\mu}(t))\|^2 \leq -\frac{\eta}{2}\|\nabla\mathcal{L}(\boldsymbol{\mu}(t))\|^2.
\tag{33}
$$

By Lemma 12 we can lower bound the gradient norm as

$$\|\nabla\mathcal{L}(\boldsymbol{\mu}(t))\| \geq \frac{\langle\nabla\mathcal{L}(\boldsymbol{\mu}(t)),\boldsymbol{\mu}(t)\rangle}{\|\boldsymbol{\mu}(t)\|} \geq \frac{\langle\nabla\mathcal{L}(\boldsymbol{\mu}(t)),\boldsymbol{\mu}(t)\rangle}{n\mu_{\max}(t)} \geq \Omega\left(\frac{\exp\left(-8U(t)\right)\pi_{\min}^2}{nd(1+\mu_{\max}(t)\sqrt{d})^2}\right)\mu_{\max}^3(t)$$

$$\overset{\text{Theorem }14}{\geq} \Omega\left(\frac{\exp\left(-8U(t)\right)\pi_{\min}^2}{nd(1+\mu_{\max}(t)\sqrt{d})^2}\right)(2\mathcal{L}(\boldsymbol{\mu}(t))^{3/2} \geq \Omega\left(\frac{\exp\left(-8U(0)\right)\pi_{\min}^2}{nd(1+\mu_{\max}(0)\sqrt{dn})^2}\right)\mathcal{L}^{3/2}(\boldsymbol{\mu}(t)).$$

$$(34)$$

Combining (34) and (33), we have

$$\mathcal{L}(\boldsymbol{\mu}(t+1)) \leq \mathcal{L}(\boldsymbol{\mu}(t)) - \frac{\eta}{2}\|\nabla\mathcal{L}(\boldsymbol{\mu}(t))\|^2 \leq \mathcal{L}(\boldsymbol{\mu}(t)) - \Omega\left(\frac{\eta\exp\left(-16U(0)\right)\pi_{\min}^4}{n^2d^2(1+\mu_{\max}(0)\sqrt{dn})^4}\right)\mathcal{L}^3(\boldsymbol{\mu}(t)).$$

$$(35)$$

Note that the above inequality implies $\mathcal{L}(\boldsymbol{\mu}(t+1)) \leq \mathcal{L}(\boldsymbol{\mu}(t))$, therefore

$$\frac{1}{\mathcal{L}^2(\boldsymbol{\mu}(t+1))} - \frac{1}{\mathcal{L}^2(\boldsymbol{\mu}(t))} = \frac{(\mathcal{L}(\boldsymbol{\mu}(t))-\mathcal{L}(\boldsymbol{\mu}(t+1)))(\mathcal{L}(\boldsymbol{\mu}(t))+\mathcal{L}(\boldsymbol{\mu}(t+1)))}{\mathcal{L}^2(\boldsymbol{\mu}(t))\mathcal{L}^2(\boldsymbol{\mu}(t+1))}$$

$$\geq \frac{(\mathcal{L}(\boldsymbol{\mu}(t))-\mathcal{L}(\boldsymbol{\mu}(t+1))\mathcal{L}(\boldsymbol{\mu}(t)))}{\mathcal{L}^4(\boldsymbol{\mu}(t))} \overset{(35)}{\geq} \Omega\left(\frac{\eta\exp\left(-16U(0)\right)\pi_{\min}^4}{n^2d^2(1+\mu_{\max}(0)\sqrt{dn})^4}\right) = \gamma.$$

On the other hand, by induction hypothesis we have $\frac{1}{\mathcal{L}^2(\boldsymbol{\mu}(t))} \geq \gamma t + \frac{1}{\mathcal{L}^2(\boldsymbol{\mu}(0))}$, combined with the above inequality, we have $\frac{1}{\mathcal{L}^2(\boldsymbol{\mu}(t+1))} \geq \frac{1}{\mathcal{L}^2(\boldsymbol{\mu}(t))} + \gamma \geq \gamma(t+1) + \frac{1}{\mathcal{L}^2(\boldsymbol{\mu}(0))}$, which finishes the proof of (30).

**Proof of** (29). The dynamics of potential function $U$ can be calculated as

$$U(\boldsymbol{\mu}(t+1)) = \sum_{i\in[n]}\|\mu_i(t+1)\|^2$$

$$= \sum_{i\in[n]}\|\mu_i(t) - \eta\nabla_{\mu_i}Q(\boldsymbol{\mu}(t)|\boldsymbol{\mu}(t))\|^2$$

$$= U(\boldsymbol{\mu}(t)) - \eta\sum_{i\in[n]}\langle\mu_i(t),\nabla_{\mu_i}Q(\boldsymbol{\mu}(t)|\boldsymbol{\mu}(t))\rangle + \eta^2\sum_{i\in[n]}\|\nabla_{\mu_i}Q(\boldsymbol{\mu}(t)|\boldsymbol{\mu}(t))\|^2 \qquad (36)$$

$$\overset{\text{Corollary }10}{=} U(\boldsymbol{\mu}(t)) - \underbrace{\eta\mathbf{E}_x\left[\|\tilde{\psi}_{\boldsymbol{\mu}(t)}(x)\|^2\right]}_{I_1} + \underbrace{\eta^2\sum_{i\in[n]}\|\nabla_{\mu_i}Q(\boldsymbol{\mu}(t)|\boldsymbol{\mu}(t))\|^2}_{I_2}.$$

By induction hypothesis, the first term $I_1$ can be bounded by Lemma 12 as

$$I_1 \geq \eta\Omega\left(\frac{\exp\left(-8U(t)\right)\pi_{\min}^2}{d(1+\mu_{\max}(t)\sqrt{d})^2}\right)\mu_{\max}^4(t) \geq \eta\Omega\left(\frac{\exp\left(-8U(0)\right)\pi_{\min}^2}{n^2d(1+\mu_{\max}(0)\sqrt{nd})^2}\right)U^2(\boldsymbol{\mu}(t)). \quad (37)$$

The second term $I_2$ is a perturbation term that can be upper bounded by Lemma 9 as

$$
\begin{aligned}
I_2 = \eta^2 \sum_{i \in [n]} \|\nabla_{\mu_i} Q(\boldsymbol{\mu}(t)|\boldsymbol{\mu}(t))\|^2 &= \eta^2 \sum_{i \in [n]} \left\| \mathbf{E}_x \left[ \psi_i(x) \sum_{k \in [n]} \psi_k(x) \mu_k(t) \right] \right\|^2 \\
&\leq \eta^2 \sum_{i \in [n]} \mathbf{E}_x \left[ \left\| \psi_i(x) \sum_{k \in [n]} \psi_k(x) \mu_k(t) \right\| \right]^2 \\
&\leq \eta^2 \sum_{i \in [n]} \mathbf{E}_x \left[ \psi_i(x) \sum_{k \in [n]} \psi_k(x) \|\mu_k(t)\| \right]^2 \\
&\leq \eta^2 \sum_{i \in [n]} \mathbf{E}_x \left[ \sqrt{\left( \sum_{k \in [n]} \psi_i^2(x)\psi_k^2(x) \right) \left( \sum_{k \in [n]} \|\mu_k(t)\|^2 \right)} \right]^2 \\
&\leq \eta^2 \sum_{i \in [n]} \mathbf{E}_x \left[ \sum_{k \in [n]} \psi_i^2(x)\psi_k^2(x) \right] \mathbf{E}_x \left[ \sum_{k \in [n]} \|\mu_k(t)\|^2 \right] \\
&= \eta^2 U(\boldsymbol{\mu}(t)) \mathbf{E}_x \left[ \sum_{i \in [n]} \sum_{k \in [n]} \psi_i^2(x)\psi_k^2(x) \right] \\
&\leq \eta^2 U(\boldsymbol{\mu}(t)) \mathbf{E}_x \left[ \left( \sum_{i \in [n]} \psi_i(x) \right) \left( \sum_{k \in [n]} \psi_k(x) \right) \right] \\
&= \eta^2 U(\boldsymbol{\mu}(t)).
\end{aligned} \tag{38}
$$

where we use triangle inequality twice at the second and third line, and Cauchy-Schwarz inequality twice at the fourth and fifth line.

Putting (38), (37) and (36) together, we get

$$
U(\boldsymbol{\mu}(t+1)) \leq U(\boldsymbol{\mu}(t)) - \eta \Omega \left( \frac{\exp\left(-8U(0)\right)\pi_{\min}^2}{n^2 d(1 + \mu_{\max}(0)\sqrt{nd})^2} \right) U^2(\boldsymbol{\mu}(t)) + \eta^2 U(\boldsymbol{\mu}(t)).
$$

Consider two cases:

a). If $\frac{U(0)}{2} \leq U(\boldsymbol{\mu}(t)) \leq U(0)$, then

$$
U(\boldsymbol{\mu}(t+1)) \leq U(\boldsymbol{\mu}(t)) - \eta U(\boldsymbol{\mu}(t)) \left( \Omega \left( \frac{\exp\left(-8U(0)\right)\pi_{\min}^2}{n^2 d(1 + \mu_{\max}(0)\sqrt{nd})^2} \right) U(\boldsymbol{\mu}(t)) - \eta \right)
$$

$$
\leq U(\boldsymbol{\mu}(t)) - \eta U(\boldsymbol{\mu}(t)) \left( \Omega \left( \frac{\exp\left(-8U(0)\right)\pi_{\min}^2}{n^2 d(1 + \mu_{\max}(0)\sqrt{nd})^2} \right) \frac{n}{2}\mu_{\max}^2(0) - \eta \right) \leq U(\boldsymbol{\mu}(t)) \leq n\mu_{\max}^2(0),
$$

note that we used $\eta \leq O\left( \frac{\exp(-8U(0))\pi_{\min}^2}{n^2 d(1 + \mu_{\max}(0)\sqrt{nd})^2} \right) \frac{n}{2}\mu_{\max}^2(0)$.

b). If $U(\boldsymbol{\mu}(t)) < \frac{1}{2}U(0)$, then $U(\boldsymbol{\mu}(t+1)) \leq (1 + \eta^2)U(\boldsymbol{\mu}(t)) \leq 2U(\boldsymbol{\mu}(t)) \leq U(0)$.

Since (29) holds in both cases, our proof is done. $\qquad \square$

## B.2 Proofs for Section 3.2

**Lemma 15.** *For any $\boldsymbol{\mu}$ satisfying $\|\mu_1\|, \|\mu_2\| \geq 10\sqrt{d}, \|\mu_3\|, \ldots, \|\mu_n\| \leq \sqrt{d}$, the gradient of $\mathcal{L}$ at $\boldsymbol{\mu}$ can be upper bounded as*

$$
\|\nabla_{\mu_i}\mathcal{L}(\boldsymbol{\mu})\| \leq 2(\|\mu_3\| + \cdots + \|\mu_n\|) + 2\exp(-d)(\|\mu_1\| + \|\mu_2\|), \forall i \in [n].
$$

*Proof.* Recall that the gradient has the form $\nabla_{\mu_i}\mathcal{L}(\boldsymbol{\mu}) = \mathbf{E}_x\left[\psi_i(x)\sum_{k\in[n]}\psi_k(x)\mu_k\right]$, hence its norm can be upper bounded as

$$
\|\nabla_{\mu_i}\mathcal{L}(\boldsymbol{\mu})\| \leq \mathbf{E}_x\left[\psi_i(x)\sum_{k\in[n]}\psi_k(x)\|\mu_k\|\right]
$$

$$
\leq \mathbf{E}_x\left[\sum_{k\in[n]}\psi_k(x)\|\mu_k\|\,\Big|\,\|x\|\leq 2\sqrt{d}\right] + \mathbf{E}_x\left[\sum_{k\in[n]}\psi_k(x)\|\mu_k\|\,\Big|\,\|x\| > 2\sqrt{d}\right]\Pr\left[\|x\| > 2\sqrt{d}\right].
$$

(39)

For any $\|x\| \leq 2\sqrt{d}$ and $i > 2$, we have $\exp(-\|x-\mu_i\|^2/2) \geq \exp(-(\|x\|+\|\mu_i\|)^2/2) \geq \exp(-9d/2)$, while for $i \in \{1,2\}$, $\exp(-\|x-\mu_i\|^2/2) \leq \exp(-(\|\mu_i\|-\|x\|)^2/2) \leq \exp(-(10\sqrt{d}-2\sqrt{d})^2/2) = \exp(-32d)$. Since $\psi_i(x) \propto \exp(-\|x-\mu_i\|^2/2)$ we have

$$
\|x\| \leq 2\sqrt{d} \Rightarrow \psi_i(x) \leq \frac{\exp(-\|x-\mu_i\|^2/2)}{\exp(-\|x-\mu_1\|^2/2)} \leq \frac{\exp(-32d)}{\exp(-9d/2)} \leq \exp(-25d), \forall i \in \{1,2\}.
$$

Therefore the first term in (36) can be bounded as $\mathbf{E}_x\left[\sum_{k\in[n]}\psi_k(x)\|\mu_k\|\,\Big|\,\|x\|\leq 2\sqrt{d}\right] \leq (\|\mu_3\| + \cdots + \|\mu_n\|) + \exp(-25d)(\|\mu_1\| + \|\mu_2\|)$.

On the other hand, by tail bound of the norm of Gaussian vectors (see Lemma 8 of [Yan et al., 2017]) we have $\Pr\left[\|x\| > 2\sqrt{d}\right] \leq \exp(-d)$. Putting everything together, (39) can be further bounded as

$$
\|\nabla_{\mu_i}\mathcal{L}(\boldsymbol{\mu})\| \leq (\|\mu_3\| + \cdots + \|\mu_n\|) + \exp(-25d)(\|\mu_1\| + \|\mu_2\|) + \exp(-d)\sum_{i\in[n]}\|\mu_i\|
$$

$$
\leq 2(\|\mu_3\| + \cdots + \|\mu_n\|) + 2\exp(-d)(\|\mu_1\| + \|\mu_2\|).
$$

$\square$

**Theorem 7.** *For any $n \geq 3$, define $\tilde{\boldsymbol{\mu}}(0) = (\mu_1^\top(0),\ldots,\mu_n^\top(0))$ as follows: $\mu_1(0) = 12\sqrt{d}e_1, \mu_2(0) = -12\sqrt{d}e_1, \mu_3(0) = \cdots = \mu_n(0) = 0$, where $e_1$ is a standard unit vector. Then population gradient EM initialized with means $\tilde{\boldsymbol{\mu}}(0)$ and equal weights $\pi_1 = \ldots = \pi_n = 1/n$ will be trapped in a bad local region around $\tilde{\boldsymbol{\mu}}(0)$ for exponentially long time $T = \frac{1}{30\eta}e^d = \frac{1}{30\eta}\exp(\Theta(U(0)))$. More rigorously, for any $0 \leq t \leq T, \exists i \in [n]$ such that*

$$
\|\mu_i(t)\| \geq 10\sqrt{d},
$$

*Proof.* We prove the following statement inductively: $\forall\, 0 \leq t \leq T$:

$$
\mu_1(t) + \mu_2(t) = 0, \mu_3(t) = \cdots = \mu_n(t) = 0 \tag{40}
$$

$$
\forall i, \|\mu_i(t) - \mu_i(0)\| \leq \eta t(60\sqrt{d}e^{-d}). \tag{41}
$$

(40) states that during the gradient EM update, $\mu_1$ will keep stationary at $0$. while the symmetry between $\mu_2,\ldots,\mu_n$ will be preserved.

The induction base is trivial. Now suppose (41), (40) holds for $0, 1, \ldots, t$, we prove the case for $t+1$.

**Proof of (40).** Due to the induction hypothesis, one can see from direct calculation that $\forall x$, we have $\psi_i(x|\boldsymbol{\mu}(t)) = \psi_i(-x|\boldsymbol{\mu}(t))$ for $i = 3,\ldots,n$, and $\psi_1(x|\boldsymbol{\mu}(t)) = \psi_2(-x|\boldsymbol{\mu}(t))$.

Consequently for $\forall i > 2$ we have

$$\nabla_{\mu_i}\mathcal{L}(\boldsymbol{\mu}(t)) = \mathbf{E}_x\left[\psi_i(x|\boldsymbol{\mu}(t))\sum_{k\in[n]}\psi_k(x|\boldsymbol{\mu}(t))\mu_k(t)\right] = \mathbf{E}_x\left[\psi_i(x)(\psi_1(x)\mu_1(t) + \psi_2(x)\mu_2(t))\right]$$

$$= \frac{1}{2}\mathbf{E}_x\left[\psi_i(x)(\psi_1(x)\mu_1(t) + \psi_2(x)\mu_2(t)) + \psi_i(-x)(\psi_1(-x)\mu_1(t) + \psi_2(-x)\mu_2(t))\right]$$

$$= \frac{1}{2}\mathbf{E}_x\left[\psi_i(x)(\psi_1(x)(\mu_1(t) + \mu_2(t)) + \psi_2(x)(\mu_2(t) + \mu_1(t)))\right] = 0 \Rightarrow \mu_1it+1) = \mu_i(t) = 0.$$

Similarly, for $\mu_1, \mu_2$ we have

$$\nabla_{\mu_1}\mathcal{L}(\boldsymbol{\mu}(t)) = \mathbf{E}_x\left[\psi_1(x|\boldsymbol{\mu}(t))\sum_{k\in[n]}\psi_k(x|\boldsymbol{\mu}(t))\mu_k(t)\right] = \mathbf{E}_x\left[\psi_1(x)(\psi_1(x)\mu_1 + \psi_2(x)\mu_2)\right]$$

$$= \mathbf{E}_x\left[\psi_2(-x)(\psi_2(-x)\mu_1 + \psi_1(-x)\mu_2)\right] = -\mathbf{E}_x\left[\psi_2(-x)(\psi_2(-x)\mu_2 + \psi_1(-x)\mu_1)\right] = -\nabla_{\mu_2}\mathcal{L}(\boldsymbol{\mu}(t)).$$

This combined with the induction hypothesis implies $\mu_2(t+1) = -\mu_1(t+1)$, (40) is proved.

**Proof of** (41)**.**

By induction hypothesis, we have $\forall i$, $\|\mu_i(t) - \mu_i(0)\| \le \eta t \cdot (60\sqrt{d}e^{-d}) \le \eta T \cdot (60\sqrt{d}e^{-d}) \le 2\sqrt{d}$. So $\forall i \in \{1,2\}, \|\mu_i(t)\| \le \|\mu_i(0)\| + 2\sqrt{d} < 15\sqrt{d}$. Then by Lemma 15, $\forall i \in [n]$ we have

$$\|\nabla_{\mu_i}\mathcal{L}(\boldsymbol{\mu}(t))\| \le 2(\|\mu_3\| + \cdots + \|\mu_n\|) + 2\exp(-d)(\|\mu_1\| + \|\mu_2\|) \le 4\exp(-d)\cdot 15\sqrt{d} = 60\sqrt{d}e^{-d},$$

note that here we used $\mu_3(t) = \cdots = \mu_n(t) = 0$. Therefore by the induction hypothesis we have $\|\mu_i(t+1) - \mu_i(0)\| \le \eta t \cdot (60\sqrt{d}e^{-d}) + \eta\|\nabla_{\mu_i}\mathcal{L}(\boldsymbol{\mu}(t))\| \le \eta(t+1) \cdot (60\sqrt{d}e^{-d})$, (41) is proven.

By (41), $\forall 0 \le t \le T$, for $i = 1, 2$ we have $\|\mu_i(t)\| \ge \|\mu_i(0)\| - \|\mu_i(t) - \mu_i(0)\| \ge 12\sqrt{d} - \eta T(60\sqrt{d}e^{-d}) \ge 12\sqrt{d} - 2\sqrt{d} = 10\sqrt{d}$. Our proof is done. $\qquad\square$

