# OpenReview forum: "Toward Global Convergence of Gradient EM for Over-Paramterized Gaussian Mixture Models"
_NeurIPS.cc/2024/Conference — NeurIPS 2024 poster_

### Official Review · Reviewer_JsCq · 2024-07-12

**Soundness:** 3
**Presentation:** 3
**Contribution:** 2
**Rating:** 6
**Confidence:** 3

**Summary:**

The paper studies the convergence of EM for learning mixtures of Gaussians.  Specifically, they consider a simplified setting where the Gaussians are in $d$-dimensions and all have covariance $I_d$.  They consider an overparameterized version of the problem where they parametrize the mixture they are trying to learn by a mixture of $n$ Gaussians with means $\mu_1, \dots , \mu_n$ and the ground truth distribution generating the data just consists of a single Gaussian $N(\mu^* , I_d)$.  The paper analyzes the dynamics of gradient EM for this problem.  The main result of the paper is proving that for this overparametrized variant, gradient EM converges to the true distribution at a rate of $1/\sqrt{t}$ with additional constants depending exponentially on the distance between the initialized means and the true mean, which they show is necessary.

There has been a long line of work on understanding the convergence of EM or gradient EM for learning mixtures of Gaussians.  Without overparametrization, provable convergence is known for mixtures of two Gaussians and it is also known that convergence fails in general for mixtures of three or more components.  For overparamterized settings, a previous work [Dwivedi et. al. 2018] shows that if we parametrize a mixture of two Gaussians and try to learn a ground truth distribution consisting of a single Gaussian, then EM converges at a $1/\sqrt{t}$ rate (as long as the mixing weights are set to be different).  This is in contrast to when we parametrize with only a single Gaussian and EM converges exponentially fast.  The results of the current paper can be seen as generalizing the results of [Dwivedi et. al. 2018] to more than two components.  The paper empirically validates their theoretical results with experiments on simple synthetic datasets.

**Strengths:**

The paper makes progress on a well-studied problem of understanding convergence of EM for learning GMMs.  They give the first global convergence results for mixtures with more than two components.

The paper overcomes nontrivial technical barriers to extend previous results to more than two components.

**Weaknesses:**

The results of the paper only work when the ground truth is "trivial" i.e. a single Gaussian.

The results are qualitatively similar to previous work on overparametrized mixtures of two Gaussians.  The contributions of the paper are mostly technical and it is a bit difficult to find a nice conceptual takeaway \--- the previous work for two components already showed that overparametrization can lead to drastically slower convergence.  It would be much more exciting and novel, say, if we could prove something when the ground truth were not just a single Gaussian.

**Questions:**

.

**Limitations:**

Yes

---

> ### Author Rebuttal · Authors · 2024-08-07
>
> Thank you for the positive review. We have addressed your concern below.
>
> > The results of the paper only work when the ground truth is "trivial" i.e. a single Gaussian.
>
> We agree that the single Gaussian ground truth is a simpler case compared to the most general problem. But our setting is nonetheless highly-nontrivial and technically challenging. In fact, even the 2-GMM problem is quite difficult with a large number of previous works (see section 2.1 for details), and our result serves as an important step towards generalizing the 2-GMM analysis to the general k-GMM learning problem.
>
> > The results are qualitatively similar to previous work on overparametrized mixtures of two Gaussians. The contributions of the paper are mostly technical and it is a bit difficult to find a nice conceptual takeaway.
>
> Our result is fundamentally different from previous works like [Dwivedi et. al. 2018]  since the mechanisms for learning general k-component GMM and 2-component GMM are different. [Dwivedi et. al. 2018] only considers 2-component GMM with *symmetric* means, and their **sub-linear convergence only happens when the mixture weights are exactly  equal (both weights being exactly $1/2$)**. On the other hand, our result applies to general GMM with arbitrary weights and asymmetric means. While the phenomenon of slow convergence observed in [Dwivedi et. al. 2018] depends on their specific setting of equal weights and symmetric means, our paper describes the general convergence behavior of k-component GMM.
>
> Another conceptual takeaway of our result, as stressed in the paper (see Section 4.1), is that one should consider the likelihood convergence rather than parametric convergence for GMM learning problem. One of our major novelty is the brand new likelihood-based framework, while previous works are mostly standard algebraic computations of the parametric convergence.

---

> > ### Comment · Reviewer_JsCq · 2024-08-11
> >
> > Thank you for the response and addressing my concerns/questions.  My overall assessment remains the same.

---

> > > ### Author Response · Authors · 2024-08-12
> > >
> > > Thank you for reading our paper and feedback! Please let us know if you have any further questions.

---

### Official Review · Reviewer_DbMw · 2024-07-13

**Soundness:** 4
**Presentation:** 4
**Contribution:** 2
**Rating:** 6
**Confidence:** 3

**Summary:**

This paper talks about the gradient-EM algorithm for over-parameterized GMM. The paper mostly shows the GLOBAL convergence and its rate when using this model to learn a single Gaussian.

**Strengths:**

I believe any non-convex global convergence optimization problem is valuable. It is an extension of Dwivedi et al. 2019.

**Weaknesses:**

1. The over-parametrized model may have severe overfitting problem.
2. The based distribution is quite easy: a single normal, with known variance. In the paper, the covariance is fixed as the identity, which simplifies the problem in a deep way. Actually for symmetric 2-GMM, there are already faster algorithms to learn both mean and cov.
3. I feel confused about the consistency and convergence in the paper. In Line 96, the convergence of KL divergence also contains the convergence of MLE, ie consistency. The convergence to the MLE is another loss function. Also in Remark 6, the convergence when sample size to infinity seems more easily ensured by WLLN.

**Questions:**

Besides the weakness above, I also have following questions:
4. If you only learn the single normal, how is the algorithm compared with Dwivedi et al. 2019 or just 2-GMM? Is it necessary to use more? Is it overfitting so the performance seems better?
5. I don’t get why the paper introduces Fact 1. It seems obvious.
6. The mean is convergent to 0 (true) instead of the MLE.

**Limitations:**

Besides above,
7. the citation format is not uniform.

---

> ### Author Rebuttal · Authors · 2024-08-07
>
> Thanks for your detailed review! We have addressed your questions below.
>
> > The over-parametrized model may have severe overfitting problem.
>
> We believe this is a misunderstanding. The aim of this paper is not to propose a new algorithm/model, but to understand the convergence behavior of the widely-used EM/gradient EM algorithm. The motivation for studying the over-parameterized regime is well-known as many have conjectured it might facilitate global convergence (see [1], [2]). Since we are considering population gradient EM, there is no overfitting in our problem setting and we leave the study of generalization theory of GMM to future works.
>
> > The based distribution is quite easy: a single normal, with known variance.
>
> This paper aims to rigorously study the optimization phenamenon in a cannonical setting.
> We agree that the base distribution is simple. However, when the learning model has more than one mixtures, the learning dynamics becomes significantly complex.
> In fact, even the further simplified setting where the learning model has only 2 mixtures and the based distribution is a single normal, the analysis in Dwivedi et al. 2019. is technically very complicated.
> The known co-variance assumption is also a standard setting widely-adopted in existing literature ([1], [2], [3]).
>
>
> > Actually for symmetric 2-GMM, there are already faster algorithms to learn both mean and cov.
>
> Again, our goal is not to design a new algorithm for learning GMM, but to understand the behavior of the widely-used EM/gradient EM algorithm. While there might be specifically designed algorithms achieving better performance for some special cases such as symmetric 2-GMM, we aim to study gradient EM as one of the most popular existing algorithms for learning general k-component GMM. We will ensure this point comes across more clearly in the final version.
>
> > I feel confused about the consistency and convergence in the paper. In Line 96, the convergence of KL divergence also contains the convergence of MLE, ie consistency. The convergence to the MLE is another loss function.
>
> While MLE and KL divergences are two different loss functions, optimizing them are actually equivalent due to the following well-known fact:
>
> $D_{KL}(p(x|\mu^*)||p(x|\mu))=-\mathrm{E}_{x\sim p(x|\mu^*)}\left[\log\left(\frac{p(x|\mu)}{p(x|\mu^*)}\right)\right]$
>
> $=-\mathrm{E}_{x\sim p(x|\mu^*)}[\log({p(x|\mu)})]$
>
> $\quad +\mathrm{E}_{x\sim p(x|\mu^*)}[\log({p(x|\mu^*)})].$
>
> Since the second term above is a constant that does not depend on $\mu$, minimizing KL is equivalent with maximizing the first term, which is just MLE. Note that this is a general fact that applies to any MLE problem, not just EM or GMM.
>
> > If you only learn the single normal, how is the algorithm compared with Dwivedi et al. 2019 or just 2-GMM? Is it necessary to use more? Is it overfitting so the performance seems better?
>
> Again, since we are considering population EM, there is no overfitting in this problem. Our algorithm is an extension of Dwivedi et al. 2019 and our main goal is not to argue that k-GMM is better or worse than 2-GMM, but to extend the previous theoretical understanding of 2-GMM to the general k-component case.
>
> > I don’t get why the paper introduces Fact 1. It seems obvious.
>
> Fact 1 implies that gradient EM is just running gradient descent on the likelihood function, an observation allowing us to introduce theoretical tools from gradient descent theory to facilitate our analysis.
>
> >  The mean is convergent to 0 (true) instead of the MLE.
>
> Since the algorithm is run on population data, there is no overfitting and the ground truth 0 is just the MLE solution.
>
> > Citation format is not uniform.
>
> Thanks for pointing it out. We will also organize our citation style and make it uniform in the revised version.
>
> References:
> - [1]. Yudong Chen, Dogyoon Song, Xumei Xi, and Yuqian Zhang. Local minima structures in gaussian mixture models. IEEE Transactions on Information Theory, 2024.
> - [2]. Chi Jin, Yuchen Zhang, Sivaraman Balakrishnan, Martin J. Wainwright, and Michael I. Jordan. Local maxima in the likelihood of gaussian mixture models: Structural results and algorithmic consequences. In Neural Information Processing Systems, 2016.
> - [3]. Sivaraman Balakrishnan, Martin J. Wainwright, and Bin Yu. Statistical guarantees for the em algorithm: From population to sample-based analysis, 2014.

---

> > ### Comment · Reviewer_DbMw · 2024-08-11
> >
> > Thank you for the addressing my concerns! Part of my questions and concerns are clarified by the authors. Although the ground truth is standard 1-d Gaussian, the work is valuable. Therefore, I would like to lift the score to 6.
> > But I still have questions about the MLE and the true (0). The author responds that the algorithm is implemented on the population data. No mentioning the possibility and reality, if the population data available, the model chosen seems inappropriate.

---

> > > ### Author Response · Authors · 2024-08-12
> > > **Thank You and Response to the Question on Population Setting**
> > >
> > > Thank you for recognizing our contribution and for raising the score!
> > >
> > > > About our choice of the population data model.
> > >
> > > We study the population setting to focus on the non-convex optimization dynamics of gradient EM algorithm. Indeed, using the population model is a standard approach in previous literature of EM analysis [1, 2], and generally in non-convex optimization [3,4].  As discussed in Remark 6, it also implies (asymptotically) the optimization convergence for reality, i.e., sample-based EM.
> > >
> > > [1]. Ji Xu, Daniel J. Hsu, and Arian Maleki. Global analysis of expectation maximization for mixtures of two gaussians. In Neural Information Processing Systems, 2016.
> > >
> > > [2]. Sivaraman Balakrishnan, Martin J. Wainwright, and Bin Yu. Statistical guarantees for the em algorithm: From population to sample-based analysis. In Annals of Statistics, 2014.
> > >
> > > [3]. Yuandong Tian. An analytical formula of population gradient for two-layered relu network and its applications in convergence and critical point analysis. In International Conference on Machine Learning, 2017.
> > >
> > > [4]. Mo Zhou, Rong Ge. A local convergence theory for mildly over-parameterized two-layer neural network. In Conference on Learning Theory, 2021.

---

### Official Review · Reviewer_DCG2 · 2024-07-13

**Soundness:** 2
**Presentation:** 2
**Contribution:** 2
**Rating:** 6
**Confidence:** 2

**Summary:**

The paper focuses on the setting of a Gaussian Mixture Model with several summands and an input vector produced by one Gaussian distribution, where it employs the Expectation-Maximization rule to infer the model's parameters. Since the problem of having arbitrary number of summands has been unsolved, the paper provides an innovative scheme which includes the computation of the likelihood function and shows that the EM algorithm converges with sublinear complexity.

The authors also show that there exist neighborhoods of slow convergence rates.

**Strengths:**

- The paper is well written, the theorems, lemmata and algorithmic steps are described gradually.
- From a first overview of the literature, the result about global convergence seems novel.
- Across section 4, there is intuition and remarks provided about the necessity of the steps.

**Weaknesses:**

- The experimental evaluation is used as a proof of concept and thus is limited. The authors could have (potentially) experimented with several datasets, with varying weights in the GMM, and try to benchmark their algorithm to compare the emergent convergence rates.

**Questions:**

NA.

**Limitations:**

NA.

---

> ### Author Rebuttal · Authors · 2024-08-07
>
> Thanks for your review and positive comment! We have addressed your question below.
>
> > The experimental evaluation is used as a proof of concept and thus is limited. The authors could have (potentially) experimented with several datasets, with varying weights in the GMM, and try to benchmark their algorithm to compare the emergent convergence rates.
>
> We note that our primary goal is to rigorously study the optimization convergence rate in a controlled setting and therefore we focus on synthetic experiments to carefully examine the phenomena and corroborate our theoretical findings.
> We also added more experiments in the rebuttal PDF (attached to the global author rebuttal for all reviewers) to verify our theoretical results:
> - Impact of mixtures weights on the convergence speed (Figure 2 Right of uploaded pdf.) We test $3$ different weight configurations of $3$-component GMM: $(\frac{1}{3}, \frac{1}{3}, \frac{1}{3})$. $(\frac{1}{6}, \frac{1}{3}, \frac{1}{2})$,$(\frac{1}{20}, \frac{1}{5}, \frac{3}{4})$. $4$ runs of each configuration are recorded, with different random initialization. Results show that the convergence is faster when the weights are more evenly distributed: the equally distributed weights of $(\frac{1}{3}, \frac{1}{3}, \frac{1}{3})$ converges with the fastest rate, while $(\frac{1}{20}, \frac{1}{5}, \frac{3}{4})$ converges the slowest.
> - Impact of initialization on the convergence speed (Figure 2 Left of uploaded pdf): We report the gradient norm in the bad initialization region constructed as counter-examples in Theorem 7. Empirically the gradient norm exponentially decreases in dimension $d$. This supports our theoretical findings that bad initialization causes exponentially slow convergence.

---

> > ### Comment · Reviewer_DCG2 · 2024-08-08
> > **Rebuttal**
> >
> > I thank the authors for their answers. They provided experiments as an attachment to their rebuttal answer. I will further study the responses to the other reviewers and update my review.

---

> > > ### Author Response · Authors · 2024-08-09
> > >
> > > Thanks for reading our response and providing feedback! We look forward to any further comments and updates.

---

### Official Review · Reviewer_6yVv · 2024-07-15

**Soundness:** 3
**Presentation:** 4
**Contribution:** 2
**Rating:** 6
**Confidence:** 4

**Summary:**

The paper considers fitting a single Gaussian with multiple-component Gaussian mixture models (GMM) through the Gradient EM algorithm. While the two balanced over-specified Gaussian setting has been widely studied in the previous work, generalizing it to multiple-component GMM requires significant algebraic efforts. The entirety of the paper is to show the $1/\sqrt{t}$ convergence rate of the population EM algorithm. In particular, the paper characterizes the explicit convergence rate of $1/\sqrt{T}$ with constants exponential in the number of components, the phenomenon that coincides with the exponential lower bound for the parameter estimation of general GMMs with no separation.

**Strengths:**

-	Extending some existing two-component results to general multiple-component GMM is non-trivial and significant. The paper nicely characterizes the convergence rate that captures some important properties of learning GMM that can be achieved by GMM.

-	The paper is well-written, emphasizing important aspects of the results and well-contrasting their techniques to existing results.

-	Proof sketch is nicely written to help readers understand their key results.

**Weaknesses:**

-	While the lower bound result (Theorem 7) is a nice addition to the literature, I believe that the gap between this lower bound and the upper bound is large, since the upper bound is exponentially slow in the number of components.

-	One important result from two specified GMM is the $n^{-1/4}$ (n is the number of samples here) statistical rate after convergence. I would like to see $n^{-1/2k}$ style results in general k-component GMM settings. At least, the authors should have discussed this aspect of previous work and contrasted the implications to k-GMM settings.

-	The experiment would have been nicer if the final statistical rates were compared.

**Questions:**

-	Maybe authors can elaborate on how their results can imply learning k-GMM with small separations?

-	In Theorem 7, there is no restriction on the step size $\eta$. I believe that the lower bound should also be able to tell that $\eta$ cannot be set too large.

-	Why only on the gradient EM? Can the analysis in the paper imply some convergence rates of the standard EM algorithm as well? I think it would make the paper much stronger if it could show that the same results hold for standard EM.

---

> ### Author Rebuttal · Authors · 2024-08-07
>
> Thank you for the detailed review. We answer each of your questions below.
>
> > The gap between this lower bound and the upper bound is large.
>
> Thank you for pointing out this problem. In the initial version we didn't optimize the exponent. Indeed, we can obtain significantly refined results which removes this gap between upper and lower bounds. The improved bounds are as follows:
>
> - [Upper bound] Consider training a student $n$-component GMM initialized from ${\mu}(0) = (\mu_1(0)^{\top},\ldots, \mu_n(0)^{\top})^{\top}$ to learn a single-component ground truth GMM $\mathcal{N}(0, I_d)$ with population gradient EM algorithm. If the step size satisfies $\eta \leq O\left(\frac{\exp\left(-8U(0)\right)\pi_{\min}^2}{n^2d^2(\frac{1}{\mu_{\max}(0)}+\mu_{\max}(0))^2}\right)$, then gradient EM converges globally with rate $$\mathcal{L}(\mu(t))\leq \frac{1}{\sqrt{\gamma t}},$$ where  $\gamma = \Omega\left(\frac{\eta\exp\left(-16U(0)\right)\pi_{\min}^4}{n^2d^2(1+\mu_{\max}(0){\sqrt{dn}})^4}\right)\in \mathrm{R}^+$. Recall that $\mu_{\max}(0)=\max\{\|\mu_1(0)\|,\ldots, \|\mu_n(0)\|\}$ and $U(0)=\sum_{i\in[n]}\|\mu_i(0)\|^2$.
>
> - [Lower Bound] For any $n\geq 3$, there exists initialization points such that, when initialized from, population gradient EM will be trapped in a bad local region around it for exponentially long time $T=\frac{1}{30\eta}\exp(\Theta(U(0)))$ as: $0\leq t\leq T$, $\exists i\in[n]$ such that $$ \|\mu_i(t)\|\geq 10\sqrt{d}. $$
>
> Both the improved lower and upper bounds are tighter than original versions since $n\mu_{\max}^2\geq U\geq \mu_{\max}^2$. Most importantly, the improved exponential factors of $\exp(\Theta(U(0)))$ in the two bounds now exactly matches, eliminating the noted large gap. Our key idea is to use $U=\sum_{i\in[n]} \|\mu_i\|^2$ (which captures information on all Gaussian means) instead of $\mu_{\max}=\max_i\{\|\mu_i\|\}$ (which reflects only  the maximum Gaussian mean) to construct our convergence rate bounds, resulting in a more fine-grained analysis and tighter bounds. We also constructed a better counter-example: ${\mu}_1(0)=12\sqrt{d}e_1, {\mu}_2(0)=-12\sqrt{d}e_1, \mu_3(0)=\cdots=\mu_n(0)=0$, where $e_1$ is a standard unit vector. This new construction applies to all $n\geq 3$ (while the original one requires $n$ being odd) and implies a tighter lower bound.
>
> With this idea, we are happy to present the **optimal** exponential factor of the convergence rate (up to a constant), which we believe might be of independent interests to future study.
>
> > Statistical rate for 2-GMM is $n^{-1/4}$. I would like to see $n^{-1/2k}$ style results in general k-component GMM settings. The authors should have discussed this.
>
>
> We agree that statistical rates for learning GMM is an important topic. However, our focus is not on this problem since we aim to study *optimization rates* rather than statistical rates of gradient EM.
> Yet we are aware that there is a long line of work on this topic. The statistical rate of $n^{-1/4}$ you noted is studied in [1] and [4]. To the best of our knowledge, the same rate for general k-GMM remains an interesting open problem and $n^{-1/2k}$ is a nice conjecture. However, our new experiments suggest that the rate seems to be still $n^{-1/4}$ for general k-GMM. (See response to the next question.) We will add these discussions and our experimental findings into the revised version.
>
> > The experiment would have been nicer if the final statistical rates were compared.
>
> Thanks for the suggestion! We have added a new experiment on this. See the global author rebuttal to all reviewers (figure 1 in attached pdf) for details. The empirical statistical rate for k-GMM is close to $n^{-1/4}$.
>
> > Maybe authors can elaborate on how their results can imply learning k-GMM with small separations?
>
> Thanks for noting this topic. Our results immediately imply that the convergence rate for learning general k-GMM with small separations will be sub-linear, since a single Gaussian is a special case of k-GMM with no separation. So the linear-contraction style analysis (such as in [2])  for well-separated GMM no longer works in this regime, and we believe our likelihood-based  framework can be helpful. We will add more corresponding discussions in the revised version.
>
>
>
> > In Theorem 7, there is no restriction on the step size.
>
> Theorem 7 applies to any positive step size $\eta$. It punishes large step sizes as $T$ scales with $1/\eta$, so the time that gradient EM gets trapped shortens as the step size increases. As long as the step size is polynomially large $\eta =poly(n,d)$, gradient EM gets trapped in the bad region for exponentially long time as $T\geq \frac{1}{poly(n,d)}\exp(\Theta(U(0)))$.
>
>
> > Why only on the gradient EM? Can the analysis in the paper imply some convergence rates of the standard EM algorithm as well?
>
> While gradient EM is equivalent with gradient descent on the likelihood function $\mathcal{L}$, standard EM can also be seen as a gradient descent-like algorithm on $\mathcal{L}$ with a coordinate-dependent step size $1/\mathrm{E}_x[\psi_i(x)]$. (See further discussions in [3]). We believe our method is also useful for standard EM. We will add more discussions on extensions to standard EM in the revised version.
>
>
>
> References:
> - [1]. Yihong Wu and Harrison H. Zhou. Randomly initialized em algorithm for two-component gaussian mixture achieves near optimality in $O(\sqrt{n})$ iterations, 2019.
> - [2]. Bowei Yan, Mingzhang Yin, and Purnamrita Sarkar. Convergence of gradient em on multi-component mixture of gaussians. Advances in Neural Information Processing Systems, 30, 2017.
> - [3]. Yudong Chen, Dogyoon Song, Xumei Xi, and Yuqian Zhang. Local minima structures in gaussian mixture models. IEEE Transactions on Information Theory, 2024.
> - [4]. Raaz Dwivedi, Nhat Ho, Koulik Khamaru, Michael I. Jordan, Martin J. Wainwright, and Bin Yu. Singularity, misspecification and the convergence rate of em. The Annals of Statistics, 2018.

---

> > ### Comment · Reviewer_6yVv · 2024-08-12
> > **Thank you for the response**
> >
> > I thank the authors for the clarification and for addressing my concerns. The additional experimental result on the statistical rate also looks interesting. I have adjusted my evaluation score accordingly.

---

### Author Rebuttal · Authors · 2024-08-07

We appreciate all the reviewers for their detailed and positive feedbacks. In the uploaded pdf file, we add several experiments:

- Experiment of statistical rates, for questions of Reviewer 6yVv (Figure 1).
- Impact of initialization on the convergence speed, for questions of Reviewer DCG2 (Figure 2, left).
- Impact of GMM mixture weights on the convergence speed, for questions of Reviewer DCG2 (Figure 2, right).

Please refer to the pdf for experimental setups and outcomes.

We have also improved our theorems, closing the gap between our upper bound and lower bound of convergence rates. This addresses the question of reviewer 6yVv. Please refer to our response to reviewer 6yVv for details.

We welcome and are happy to answer any further questions from reviewers.

---

### Comment · Area_Chair_7pB2 · 2024-08-08

Dear authors, dear reviewers,

the discussion period has begun as the authors have provided their rebuttals.
I encourage the reviewers to read all the reviews and the corresponding rebuttals: the current period might be an opportunity for further clarification on the paper results and in general to engage in an open and constructive exchange.

Many thanks for your work.
The AC

---

### Decision · Program_Chairs · 2024-09-25

**Decision:**

Accept (poster)

**Comment:**

As stated in the abstract, the paper's focus is the gradient expectation-maximization algorithm for Gaussian mixture models in the overparametrized regime: the authors consider the convergence properties of the problem when a single Gaussian is fitted by a mixture, showing that the considered algorithm converges globally with sublinear rate. The paper collected a positive consensus due to its clarity and novelty. Despite the apparent simplicity of the set-up, the extension of results already available in the literature to the case of generic Gaussian mixture is interesting, as it requires the introduction and application of nontrivial technical steps for the study of the non-convex optimization dynamics of the algorithm. I therefore recommend the acceptance of the manuscript.